# Structure and genome editing of type I-B CRISPR-Cas

Meiling Lu[1,2,9] ✉, Chenlin Yu[1,9], Yuwen Zhang[1], Wenjun Ju[1], Zhi Ye[1], Chenyang Hua[1], Jinze Mao[3], Chunyi Hu [4,5], Zhenhuang Yang[6] ✉ & Yibei Xiao [2,7,8] ✉

Type I CRISPR-Cas systems employ multi-subunit effector Cascade and helicase-nuclease Cas3 to target and degrade foreign nucleic acids, representing the most abundant RNA-guided adaptive immune systems in prokaryotes. Their ability to cause long fragment deletions have led to increasing interests in eukaryotic genome editing. While the Cascade structures of all other six type I systems have been determined, the structure of the most evolutionarily conserved type I-B Cascade is still missing. Here, we present two cryo-EM structures of the *Synechocystis sp*. PCC 6714 (*Syn*) type I-B Cascade, revealing the molecular mechanisms that underlie RNA-directed Cascade assembly, target DNA recognition, and local conformational changes of the effector complex upon R-loop formation. Remarkably, a loop of Cas5 directly intercalated into the major groove of the PAM and facilitated PAM recognition. We further characterized the genome editing profiles of this I-B Cascade-Cas3 in human CD3[+] T cells using mRNA-mediated delivery, which led to unidirectional 4.5 kb deletion in *TRAC* locus and achieved an editing efficiency up to 41.2%. Our study provides the structural basis for understanding target DNA recognition by type I-B Cascade and lays foundation for harnessing this system for long range genome editing in human T cells.

CRISPR-Cas system is an adaptive immune system that defends prokaryotes against the invasion of foreign genetic elements[1–3]. In these systems, CRISPR-associated (Cas) protein(s) assemble with transcribed and processed CRISPR RNAs (crRNAs) to form the effector complex that degrades the complementary invading nucleic acid[4–6]. Based on the constitute of the effector, the CRISPR-Cas systems are broadly divided into two classes: Class 1 utilizes multi-subunit effector proteins and Class 2 employs only a single effector protein[7]. Class 2 CRISPR-Cas systems are well studied and their effectors, including Cas9 and Cas12, have been widely employed for genome editing[8,9]. However, Class 2 systems only account for about 10% of all discovered CRISPR-Cas systems[10], the more prevalent Class 1 CRISPR-Cas systems are a huge reservoir of potential genome manipulation tools that await further exploration.

Type I CRISPR-Cas systems are the most abundant Class 1 systems[7], and are further divided into seven types I-A through I-G.

[1]Department of Biochemistry, School of Life Science and Technology, China Pharmaceutical University, Nanjing 211198, China. [2]State Key Laboratory of Natural Medicines, China Pharmaceutical University, Nanjing 211198, China. [3]Nanjing Foreign Language School, Nanjing 210008, China. [4]Department of Biological Sciences, Faculty of Science, National University of Singapore, Singapore 117543, Singapore. [5]Precision Medicine Translational Research Programme (TRP), Department of Biochemistry, Yong Loo Lin School of Medicine, National University of Singapore, Singapore 117543, Singapore. [6]Institute for Hepatology, National Clinical Research Center for Infectious Disease, Shenzhen Third People's Hospital, Shenzhen, Guangdong 518112, China. [7]Department of Pharmacology, School of Pharmacy, China Pharmaceutical University, Nanjing 211198, China. [8]Chongqing Innovation Institute of China Pharmaceutical University, Chongqing 401135, China. [9]These authors contributed equally: Meiling Lu, Chenlin Yu. ✉e-mail: lumeiling@cpu.edu.cn; yanginchina@hotmail.com; yibei.xiao@cpu.edu.cn

These systems are characterized by the coordinated action of Cascade (CRISPR-associated complex for antiviral defense) which binds complementary target dsDNA and a nuclease-helicase subunit Cas3 for processive DNA degradation[7,11]. All type I-A[12], I-C[13–15], I-D[16,17], I-E[18–20], I-F[21–23], I-G[24] Cascade structures have been determined to date, except for type I-B Cascade. A typical Cascade contains a large subunit that detects the protospacer adjacent motif (PAM) on the target DNA, a belly composed of two to five copies of small subunits, a backbone for crRNA binding, a crRNA-processing nuclease, and a single-copy subunit flanked to the 5′end of the crRNA[25]. In type, I-C[13–15], I-D[16,17,26], I-E[18–20], and I-F[21–23] systems, Cas3 recruitment is dependent on R-loop formation upon target DNA recognition by Cascade. By contrast, Cas3 of I-A[12] and I-G[24] subtypes is a stable component of the complex effector even in the absence of target DNA. The ability to cause long fragment deletions has been validated in microbes[27,28], plants[29,30], and mammalian cells[31–33]. Cascade or Cas3 fused with other regulatory proteins or modifiers have been proven to regulate gene expression or induce random mutagenesis[34,35].

Several studies have suggested potential applications using type I-B system. For example, endogenous type I-B systems have been redirected for gene deletion/ insertion in several native hosts[36–38], and the reconstructed type I-B interference machinery can scavenge target genes on plasmids[36,39,40]. Recently, a type I-B CRISPR-associated transposase (CAST) system utilizes I-B Cascade to target DNA and recruits Tn7-like transposase for achieving site-specific gene insertion has been reported[41]. However, structural-based mechanistic understanding and validation of gene editing capabilities of type I-B CRISPR-Cas3 system are still not fully understood.

In this study, we reconstitute a *Syn* type I-B Cascade and delineate its broader PAM requirement with the optimal preference as 5′-A-Y-G-3′. We determine the cryo-EM structures of *Syn* type I-B Cascade bound to the dsDNA target and show interactions for PAM recognition, NTS stabilization, and conformational changes of the large subunit in two functional states. Uniquely, both Cas5b and Cas8b subunits are involved in PAM recognition. In addition, we introduce the *Syn* type I-B system into human CD3[+] T cells using mRNA delivery, which achieved a satisfactory editing efficiency of up to 41.2% and unidirectional 4.5 kb deletion. Our results form the structural basis for understanding target DNA recognition mechanisms by type I-B Cascade, filling the last missing piece of type I Cascade, and show the potential of type I-B CRISPR-Cas3 in large genome fragment deletion in T-cell engineering.

## Results

### Reconstitution and PAM determination of type I-B system
*Synechocystis sp.* PCC 6714 encodes a type I-B CRISPR locus which was subdivided as Myxan based on the properties of its large subunit[42] (Fig. 1A). The initial nomenclature of the large subunit as Cmx8 was updated to Cas8b in 2020 by Makarova et al.[7]. Like its counterparts in I-C[13–15] and I-D[16,17] systems, the *cas8b* large subunit of this system also includes an internal ribosome-binding site at its 3′ terminus, which encodes a separate small subunit Cas11[17]. To recapitulate this type I-B system, plasmids encoding Cas8b, Cas7b, Cas5b, Cas6b, and Cas11b were co-expressed with associated CRISPR array, in *E. coli*, and co-purified as an assembled complex through a Strep-tag fused to the N-terminus of Cas8b. Size-exclusion chromatography of the affinity-purified sample indicated successful assembly of a type I-B Cascade, which was eluted at a volume corresponding to slightly smaller than 440 kDa (Fig. 1B). SDS-PAGE revealed the presence of Cas8b (70 kDa), Cas7b (35 kDa), Cas5b (26.5 kDa), Cas6b (24.4 kDa), as well as an expected Cas11b at <15 kDa (Fig. 1C). A ~ 71 nucleotides long crRNA was co-purified, consistent with the length of a full spacer-repeat crRNA unit (Fig. 1D).

PAM-dependent recognition forms the basis to distinguish host DNA from foreign nucleic acids in type I CRISPR immunity[6,43]. To identify the optimal PAM sequence of the *Syn* type I-B system, pET28a-NNN-protospacer plasmid libraries were constructed to generate potential PAM sequence variety. All NNN-protospacers were then amplified into 161 bp 6-FAM-labeled dsDNA and incubated with the *Syn*Cascade for Electrophoretic Mobility Shift Assay (EMSA). Specific bands, indicative of DNA binding with low *Syn*Cascade complex concentrations, were singled out and followed with Sanger sequencing to map the PAM preference. The sequencing results of the recovered DNA suggested that the −3 position of PAM showed a strong bias for adenine (A), while the −2 and −1 positions preferred Y and G, respectively (Fig. 1E). To further validate the propensity of nucleotides for the −2 and −1 positions, we analyzed the binding affinity of the 16 ANN-protospacer substrates with *Syn*Cascade. With the −3 site of PAM fixed as A, *Syn*Cascade prefers to bind with DNA substrate containing 5′-A-Y-N-3′ PAM sequence than 5′-A-R-N-3′ (Supplementary Fig. 1A). Comparing the binding affinities of DNA sequences containing 5′-A-Y-N-3′ with the Cascade complex revealed that AYG has the greatest binding capacity. ATA was a close second, with marginally lower affinity. Both ATY and ACM displayed moderate binding affinities, whereas ACT shows the lowest preference within the AYN group (Supplementary Fig. 1B). The 5′-A-Y-G-3′ preference was re-validated by binding assays between Cascade and eight 5′-N-Y-G-3′ protospacers (Supplementary Fig. 1C, D). The results showed that the nucleotide preference for the −3 position in the PAM sequence was indeed adenine. Taken together, our results validate that the reconstituted *Syn* type I-B Cascade appears to have a broader PAM requirement with the best preference as 5′-A-Y-G −3′.

### Integral type I-B Cascade-DNA assembly and Cryo-EM structure analysis
To elucidate the detailed mechanism of how the *Syn* type I-B Cascade recognizes target DNA, we incubated a 59-bp dsDNA target containing an ATG-PAM (Fig. 2A) with *Syn*Cascade in a 1:3 molar ratio at 25 °C for 1 h, the unbound dsDNA was then removed by SEC purification. Thereafter, cryo-EM was employed to reveal the structural features of the integral complex. Raw micrographs and reference-free 2D class averages clearly showed particles with a "sea horse"-like shape (Supplementary Fig. 2). The final 3D reconstruction reached an overall resolution higher than 3.6 Å, which was sufficient to identify the direction of the main chain and the clear side chains (Fig. 2B). Some periphery regions including the Cas6b/crRNA 3′-hairpin were either not well resolved or were too degenerate for modeling. The target strand (TS) is embedded within the complex and hybridized with the crRNA, representing the full R-loop formed state. However, the non-target strand (NTS) was not fully visible, particularly the bulge for Cas3 recruitment (Fig. 2D). We also observed a subset of particles within our cryo-EM dataset that formed a partial R-loop state, with only 5 nt of the TS hybridized to the crRNA, alongside duplex DNA bound with Cas8b (Fig. 2C, E). This led to the discovery of an additional structure, resolved at 3.8 Å (Supplementary Fig. 2).

The stoichiometry of *Syn*Cascade was Cas8b$_1$-Cas7b$_7$-Cas5b$_1$-Cas6b$_1$-Cas11b$_3$. The helical backbone of the complex, composed of seven successive Cas7b subunits, was clearly identified in the density. It shared structural similarities with that of I-A[12], I-C[13–15], and I-D[16,17] Cascades, featuring a longer helical backbone. The crRNA bound to the backbone and threaded through the "finger" domains (Fig. 2F). Each Cas7b subunit occupied 6 nucleotides of the crRNA with a recurring periodic pattern of 5 + 1 nt, where the sixth base flipped out in the opposite direction to the other five[44] (Fig. 2G). The Cas5b subunit is located at the top of the complex and recognizes the 5′ handle of crRNA.

Adjacent to Cas5b is the large subunit Cas8b, which displays very low sequence similarity with the large subunits of any other known type I CRISPR systems. Meanwhile, it has low sequence identity with

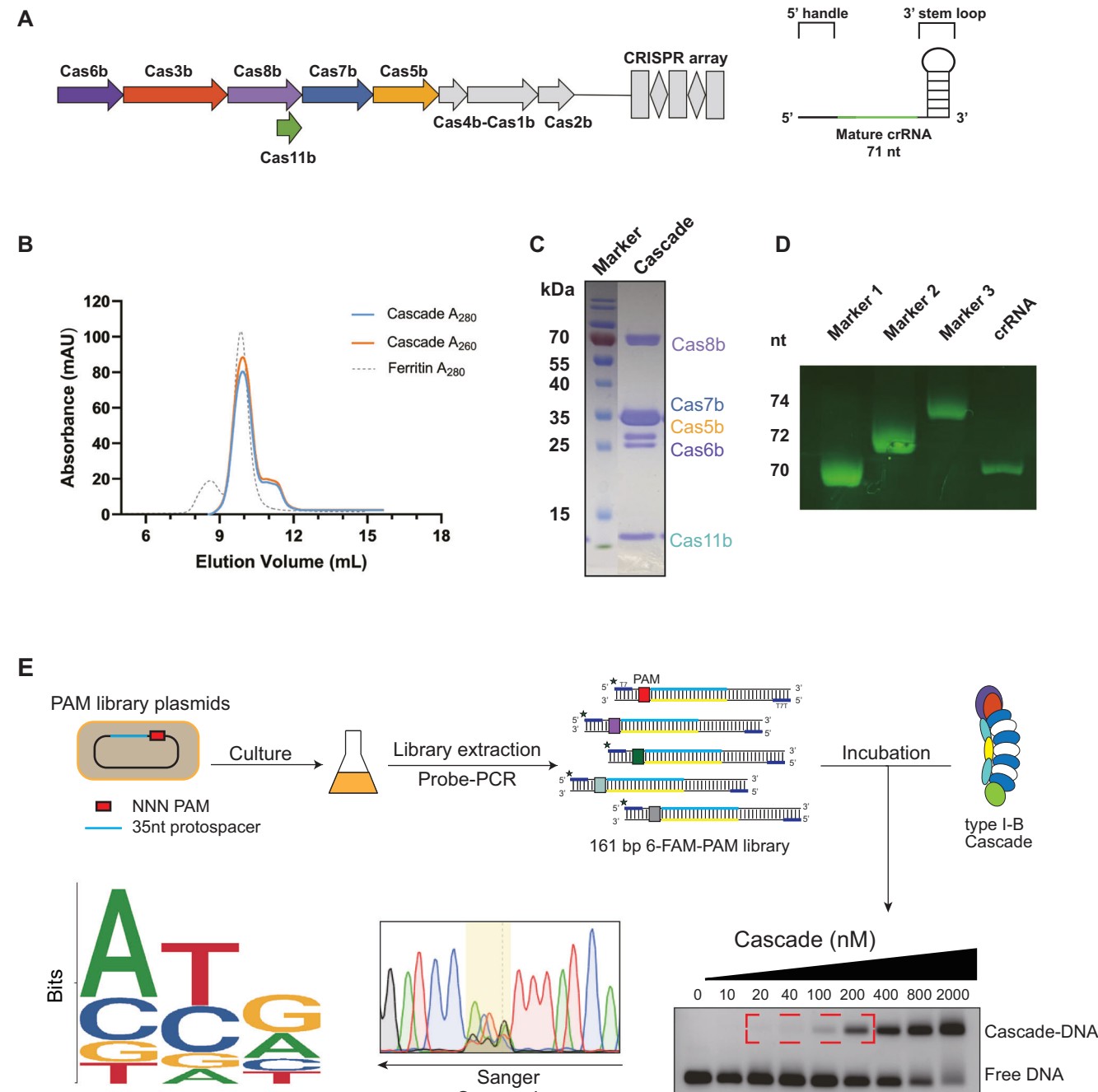

**Fig. 1 | Heterologous expression and PAM sequence determination of type I-B Cascade. A** Schematic of *Synechocystis sp*. PCC 6714 type I-B operon (left) and mature crRNA (right). **B** The SEC chromatogram of the recombinant Cascade was purified by an N-terminal Strep-tag on Cas8b, and the Ferritin (Mr 440,000) was used as the standard. **C** Coomassie blue stained SDS-PAGE gel of Cascade sample collected at the main elution peak of Cascade in **B**. **D** Ethidium bromide stained Urea-PAGE gel of the crRNA isolated from the same sample as **C**, and three transcribed RNA of different lengths were used as the markers. **E** Experimental strategy for PAM identification using plasmid libraries and fluorescent-labeled dsDNA to trap the candidates. To avoid the non-specific binding, only the DNA hits bound with Cascade of low concentrations were recovered (lane 3–6) as a template for further PCR amplification, individually. The weblog for the trinucleotide PAM consensus observed by Sanger sequencing results, with the PAM motif regions highlighted in the yellow box.

the large subunit of other subtypes of I-B system, e.g., only 29.85% similarity with the Cas8 of the I-B CAST system (Supplementary Fig. 3). The Cas8b large subunit comprises an N-terminal domain and a helical C-terminal domain. In contrast to other type I systems, where the NTD of the large subunit displays poor density and cannot be accurately modeled in the partial R-loop state, the Cas8b NTD in the *Syn* type I-B Cascade structure is well resolved in both full and partial R-loop states. The only exception is the β-sheet consisting of residues 94–117, which

is positioned laterally to the main structure (Supplementary Fig. 4), and is missing in the map. This β-sheet is located in a position similar to the recruit loop of Cas8b in the *Nla* I-C subtype, which may function in Cas3 recruitment[15]. The C-terminal portion, identical to the Cas11b small subunit, is similar in size and secondary structure of the α-helical bundle observed in the small Cas11 subunit in most Class I effectors, though it exhibits low identity with other Cas11 proteins[16]. The C-terminal domain of Cas8b and three Cas11b small subunits together

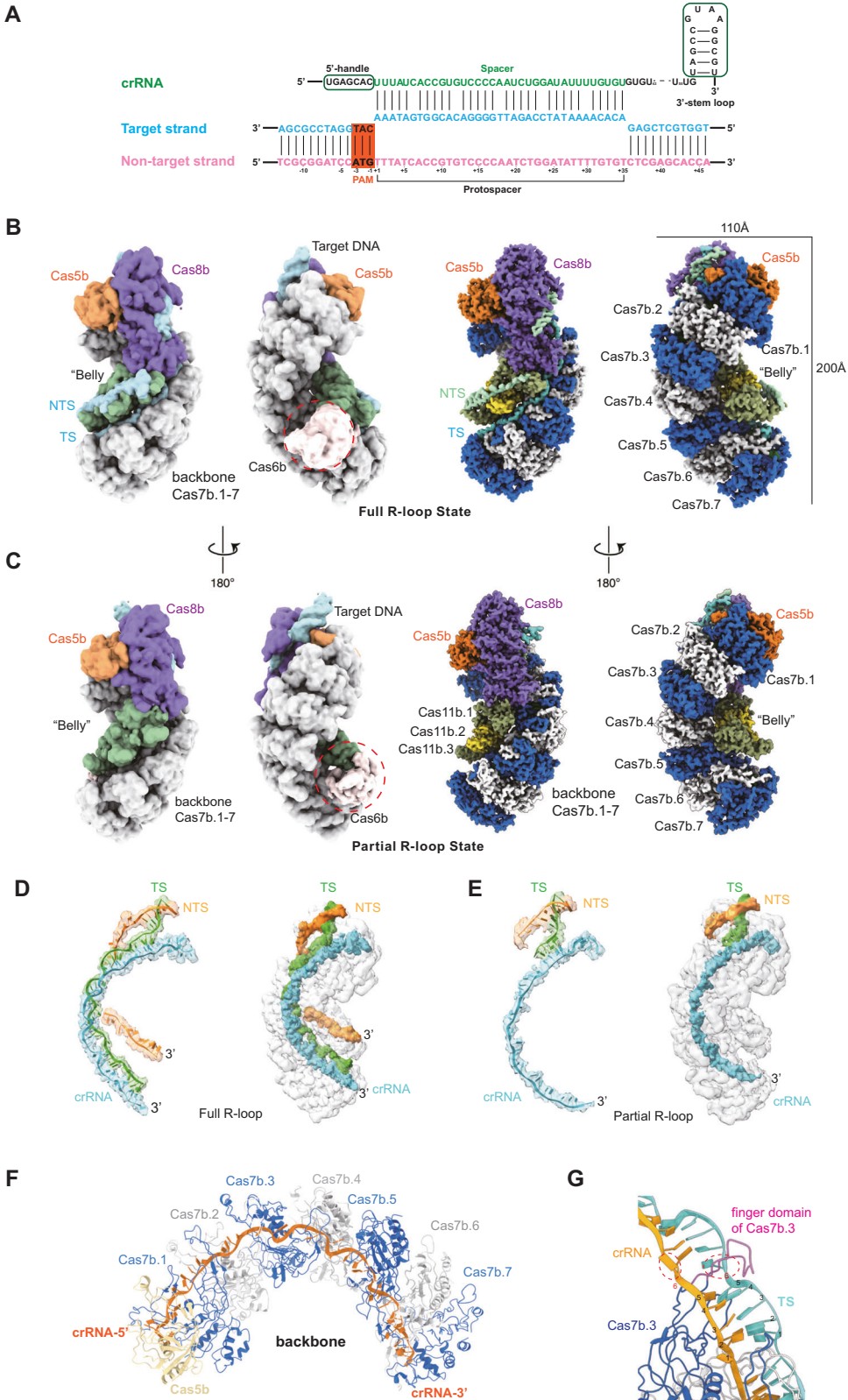

**Fig. 2 | Cryo-EM snapshots of the type I-B Cascade-dsDNA complex. A** crRNA and dsDNA sequences were used to program type I-B Cascade for structure studies. Residue numbers and color schemes are followed throughout the text. The PAM region is highlighted in the orange box. **B** Schematics of the cryo-EM density (left) and modeled structure (right) of full R-loop formation state. **C** Schematics of the cryo-EM density (left) and modeled structure (right) of partial R-loop formation state. **D** Cryo-EM density of nucleic acids in full R-loop state. **E** Cryo-EM density of nucleic acids in partial R-loop state. **F** The representation of Cas7b_7 backbone binding with crRNA is shown in the cartoon. **G** The finger domain of the Cas7b subunit disrupts the complementary base pairing between crRNA and target DNA strand at intervals of 5 bases.

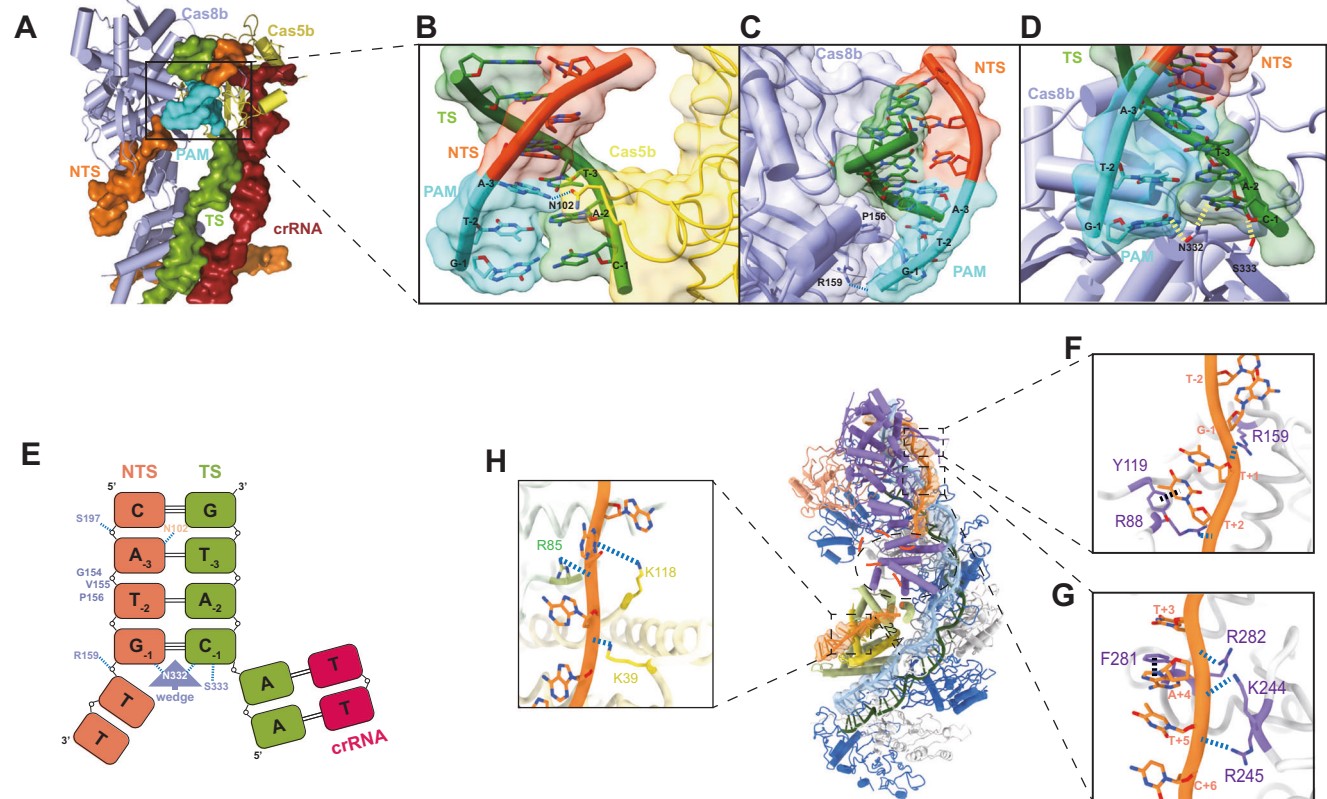

**Fig. 3 | Structural basis for PAM recognition and NTS stabilization. A** Binding pattern of target DNA with Cas8b and Cas5b. **B** N102 of Cas5b formed a H-bond with $A_{NT-3}$ (blue dash lines). **C** A "GVP" loop (154–156) of Cas8b is proximal to and interacts with the minor groove of the PAM duplex. **D** The wedge N332 of Cas8b forms two H-bonds with $G_{NT-1}$ and $C_{T-1}$, respectively. The adjacent S333 interacted with the ribose of $C_{T-1}$. H-bonds are depicted using yellow dashed lines. **E** Schematic

of the residues involved in PAM recognition for *Syn*Cascade. **F, G** Specific residues in Cas8b NTD involved in NTS stabilization. The positively charged residues and aromatic residues form non-specific interactions with the NTS backbone and bases, respectively. H-bond and aromatic forces are illustrated using blue and black dashed lines, respectively. **H** Specific residues in Cas11b small subunits involved in NTS stabilization. H-bonds are depicted using blue dashed lines.

formed the inner "belly" of the integral complex. They were tasked with providing support for the non-target strand (NTS) within the full R-loop state structure (explained in detail later).

### Cas8b NTD and Cas5b are responsible for PAM recognition

In type I systems, the best-studied PAM recognition typically involves large subunit-mediated DNA minor groove contacts. This recognition involves three components: the specific residues on a Gly-rich loop in the large subunit's NTD that interacts with the DNA's minor groove, a Gln-wedge that inserts itself into the dsDNA path beneath the PAM, and a Lys-finger that favorably forms electrostatic interactions with a pyrimidine in the PAM[45]. However, the Gln-wedge may change to an Asn-wedge in type I-C[14] or to a Lys-wedge in type I-F[23], and the Lys-finger is replaced by Asn in type I-C[14] and type I-F[23].

In our I-B system, a loop comprised of residues 154–156 (GVP), bridging two helices in the NTD of *Syn*Cas8b, is proximal to and interacts with the minor groove of the PAM duplex. Notably, a "GNS" loop (residues 101–103) of *Syn*Cas5b intercalated into the major groove of the PAM, opposite to the "GVP" loop from Cas8b, aiding in PAM recognition (Fig. 3A, C). Within the "GNS" loop, the N102 residue closely interacts with the amino group of $A_{NT-3}$, forming a hydrogen bond (Fig. 3B). This might explain why *Syn* type I-B Cascade strongly prefers the PAM-3 as A. Similarly, the G101 residue is also adjacent to the major groove, allowing ample room for $A_{T-2}$. In the type I-D system, the Cas5d subunit also closely contacts the major groove of the target DNA, but it merely serves an accessory role in stabilizing the DNA[15]. Compared to the wild type (WT), the G101A and N102A mutants of *Syn*Cas5b significantly reduced the DNA binding affinity

and preference for the PAM sequence (Supplementary Fig. 5). This suggests that the "GNS" loop of *Syn*Cas5b plays a pivotal role in PAM recognition.

Residue P156 of "GVP" motif in Cas8b is situated close to the minor groove of the PAM sequence (Fig. 3C). Its rigidity may elucidate why PAM-2 favors Y: a purine nucleotide would cause a steric clash with the "GVP" loop (Supplementary Fig. 6). The insertion of a wedge structure to initiate DNA duplex unwinding was also observed in *Syn*Cas8b. Within this wedge, N332 establishes two hydrogen bonds (Fig. 3D): one with the N1 of $G_{NT-1}$ and the other with the N3 of $C_{T-1}$, making PAM-1 more favorable to G. Adjacently, S333 forms a hydrogen bond with the oxygen of $C_{T-1}$'s ribose (Fig. 3D). These interactions, along with other backbone-stabilizing interactions, aid in binding to dsDNA targets. Taken together, these results suggest that both the "GNS" loop of Cas5b and the "GVP" loop of Cas8b are responsible for AYG-PAM recognition in *Syn* type I-B Cascade (Fig. 3E).

### NTS stabilization and conformational dynamics during full R-loop formation

In our full R-loop state structure, we modeled 6 bp of PAM-proximal dsDNA and 19 nt of NTS ssDNA. Out of these, 9 nt of NTS ssDNA were directly located downstream of PAM, while the other 10 nt constituted the PAM-distal region stabilized by the small subunits were modeled using "A" (Fig. 2D). Positively charged residues (R88, R159, K244, R245 and R282) within the Cas8b NTD made sequence-independent contacts with the negatively charged NTS backbone. In addition to electrostatic contacts, we identified aromatic residues (Y119 and F281) of Cas8b NTD that participated in the stacking interactions with

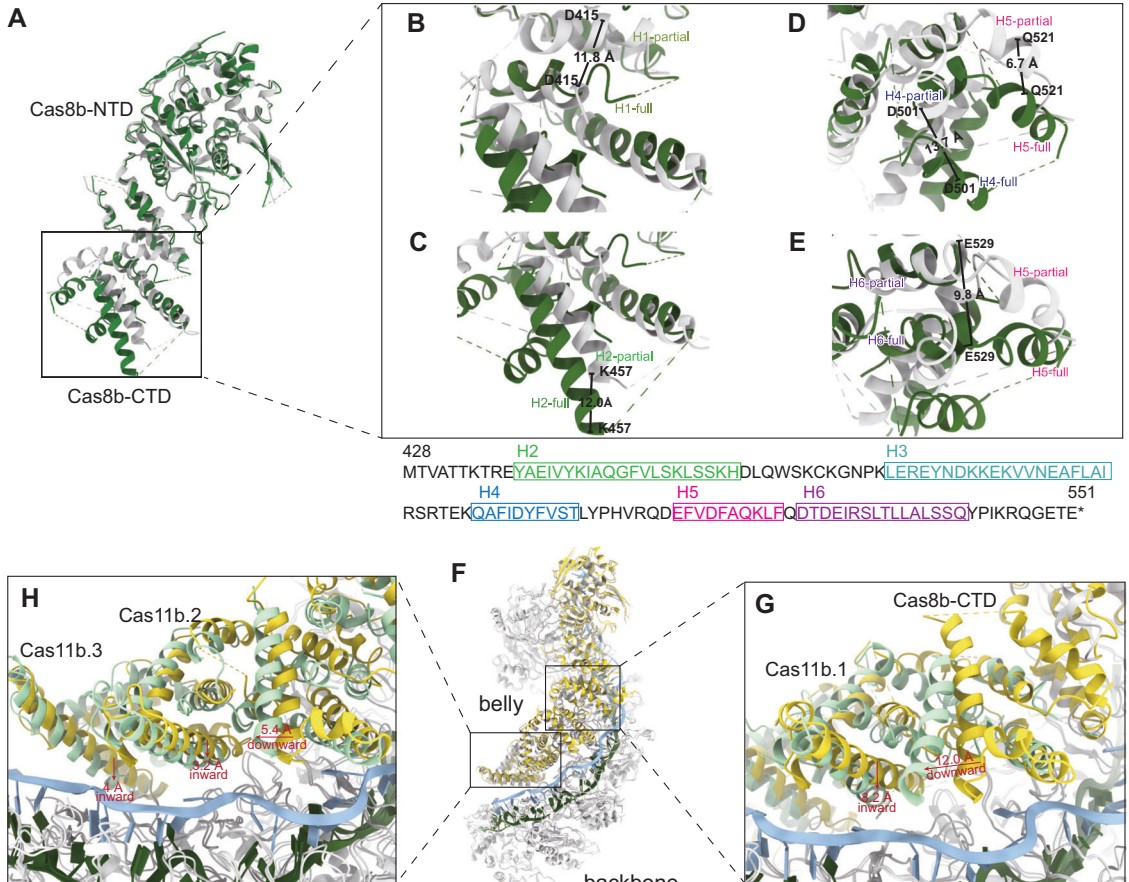

**Fig. 4 | Conformational changes during full R-loop formation. A** Alignment of Cas8b in two states. **B–E** Conformational changes in Cas8b-CTD of *Syn*Cascade between full (green) and partial (gray) R-loop state. **F** Alignment of two Cascade structures with the backbones overlapped. **G, H** Conformational transition in the belly (Cas8b CTD and Cas11-3) between full (cyan) and partial (yellow) R-loop state. Arrows indicate the direction of movements upon full R-loop formation.

NTS bases, which stabilizing the single-stranded region of the NTS seed (Fig. 3F, G). After modeling 9 nucleotides downstream of the PAM position, the density of the NTS located on the surface of the tail of Cas8b NTD deteriorated in an unclear direction. Similar stimulation was observed in most other type I systems except for I-C systems[14,15].

In our partial R-loop state structure, only 5 nt of the TS hybridizing with the crRNA and 3 nt of the NTS downstream of the PAM were modeled (Fig. 2E). Comparative analysis of the overall structure of Cas8b in the two different states showed that its CTD extended in the full R-loop structure (Fig. 4A), with an RMSD of 6.7 Å. This extension was accompanied by a significant downward and outward displacement of 11.8 Å, 12.0 Å, 13.7 Å, 6.7 Å, and 9.8 Å in Helix1, Helix2, Helix4, Helix5, and Helix6, respectively (Fig. 4B–E). While the position of Helix3 is unchanged. This extension might promote the creation of the ssDNA-binding groove, which is vital for the NTS supporting and Cas3 recruitment.

Comparison of the "belly" in the two structures reveals a pivoting motion of the extended Cas8b CTD, accompanied by a correlated motion and rotation of the three Cas11b subunits reflected by an RMSD of 3.15 Å (Fig. 4F–H). This motion leads to a reduction in the spatial distance between the belly and the backbone. Concurrently, the small subunits position the NTS ~ 22 Å above the DNA/crRNA heteroduplex, facilitated by electrostatic interactions between the positively charged residues (R85, K118, K39) and the negatively charged backbone of the NTS (Fig. 3H). These features suggest that the R-loop formation follows a kinetically favorable mechanism, analogous to that observed in type I-C[14].

## Type I-B CRISPR-Cas3 mediated genome editing in human cells
CRISPR/Cas9 engineered T cells showed high efficiency and safety in cancer immunotherapy[46–49], we therefore explored the potential usage of *Syn*CRISPR-Cas3 for genome editing in human cells by disrupting T-cell receptor α constant (*TRAC*) locus in CD3+ T cells. Previous studies showed that the *Nla* I-C system achieved a 95% editing efficiency using RNP delivery, but resulted in only an 8% lesion of *EGFP* in HAP1 cells when using mRNA delivery[32]. This suggests that mRNA-mediated delivery might not be optimally effective, but we posited that the efficacy of mRNA delivery could vary depending on the specific CRISPR-Cas system and cell types.

Hence, we examined the editing efficiency of this type I-B Cascade-Cas3 system in an mRNA-mediated manner. Two 71-nt crRNAs containing protospacer G1 or G2 that target a 35-bp region in *TRAC* locus flanked by a 5′-ATG-3′ PAM were designed (Fig. 5A). 5′capped and 3′polyA-tailed mRNAs for *cas3, cas8b, cas7b, cas5b, cas6b* and *cas11b* of *Syn* type I-B system were transcribed in vitro and then electroporated into CD3+ T cells, along with mature crRNA (Fig. 5B). The endogenous TCR-α chain was disrupted by knockout of *TRAC* gene, and a specific monoclonal antibody was utilized to track TCR-αβ expression, which only occurs on the T-cell surface when both TCR-α and -β chains are co-expressed. The editing efficiency was monitored using flow cytometry based on the expression level of TCR. *cas* mRNAs and a crRNA targeting a non-*TRAC* gene were delivered as a negative control, resulting in a negligible signal above the untreated background. It is shown that an average of 35.56% and 36.62% editing was generated by crRNA containing G1 and G2, respectively (Fig. 5C, D). The cells with significantly reduced TCR expression were collected by flow cytometry, and their

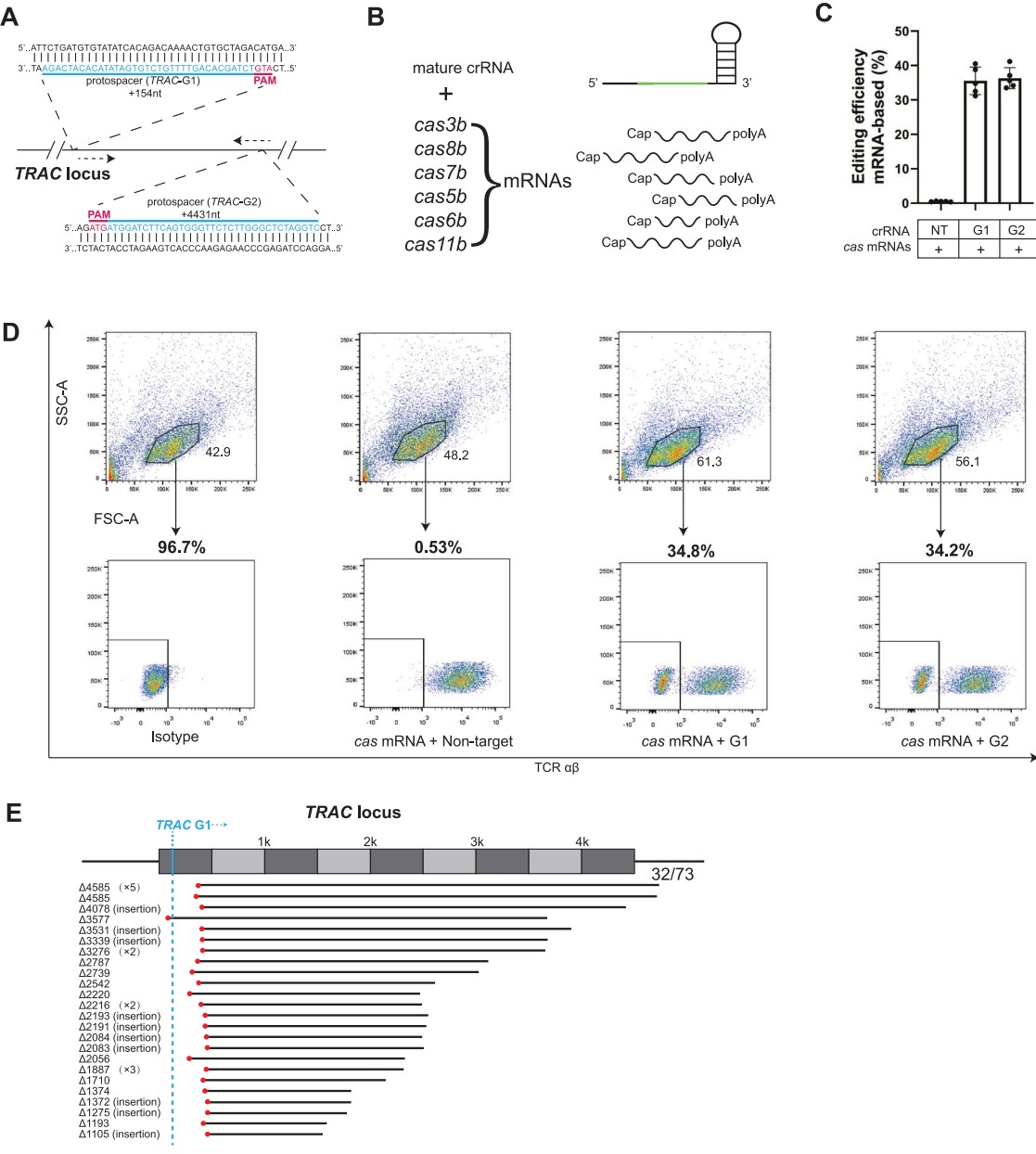

**Fig. 5 | mRNA delivery of *Syn*CRISPR-Cas operon achieves high-efficiency genome editing in CD3⁺ T cells. A** Schematic of the *TRAC* locus, with protospacers for the two TRAC-targeting Cascades shown in blue and corresponding PAM in magenta. **B** Schematics of *cas* mRNAs and mature crRNA used in **C**. The *TRAC*-targeting CRISPR spacer is shown in green. **C** mRNAs encoding *Syn* proteins were electroporated into T cells, along with a TRAC-targeting CRISPR in the form of mature crRNA. Editing efficiencies were evaluated and plotted. Data are shown as mean ± SEM, *n* = 5 independent healthy donors. **D** Representative flow cytometry plots of experiment in **C**, with percentages of TCR- T cells in the population shown on the top. **E** 32 long-range deletions (>1 kb) location at the *TRAC* locus for G1, revealed by PacBio sequencing of the long-range PCR products using the extracted genomic DNA as a template. Deleted genomic regions, G1-protospacer, and the onset of deletion are shown as black lines, blue lines, and red dots, respectively.

genomic DNAs were extracted to further characterize the *Syn*Cascade-Cas3 genome editing profile. Long-range PCR was performed using the extracted genomic DNAs as templates, and a total of 73 deletion fragments which were all generated by G1 were obtained after PacBio sequencing. Among them, 32 fragments with deletion lengths larger than 1 kb were shown in Fig. 5E, the remaining were listed in Table S5. The deletions were uniformly initiated at a frame from +320 nt to +348 nt of the PAM, except for one that started at −85 nt of the PAM. The deletion endpoints varied within the -4.5 kb PAM-proximal region (Fig. 5E). Taken together, we concluded that *Syn*CRISPR-Cas3 could be efficiently delivered using mRNA and create a long-spectrum deletion that is unidirectional relative to the target in human T cells.

## Discussion

The type I CRISPR-Cas system targets invasive genetic elements during the interference stage in a stepwise manner[7,11], which may minimize off-target effects and enable long-range deletions that other CRISPR-Cas systems cannot achieve[50–52]. With the elucidation of the type I-B Cascade structure in this study, now we have a complete picture of all seven type I CRISPR Cascades. As shown in Fig. 6, the structural and functional features of the Cascade "backbone" in the type I CRISPR-Cas system are similar, albeit with slight differences in the number of subunits and the degree of bending. Coincidentally, the number of copies of the small subunit, including the CTD of the large subunit that comprises the Cascade "belly",

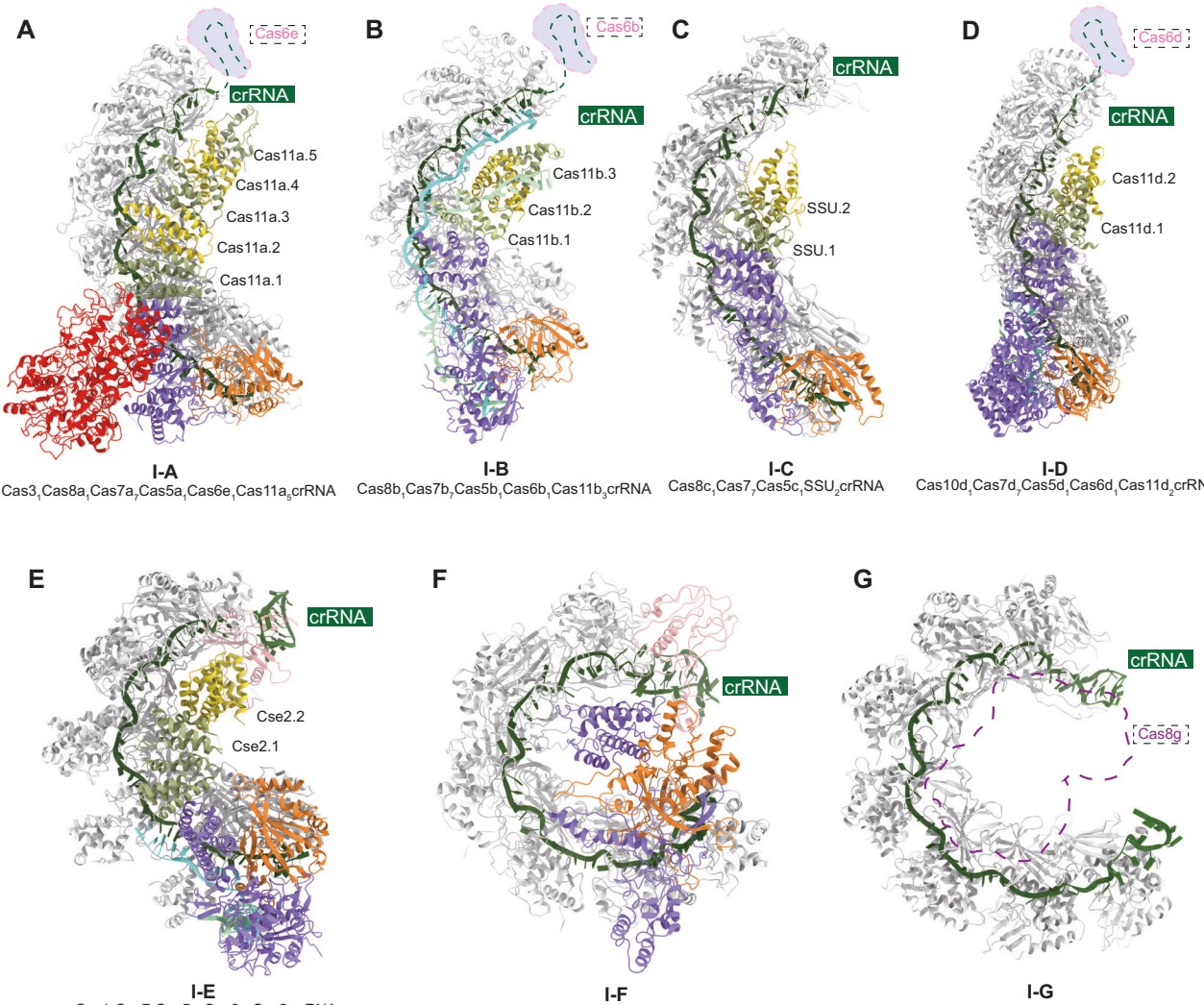

**Fig. 6 | Comparison of type I CRISPR effector structures.** The Cas7 backbone is shown in light gray, Cas5 in orange, Cas6 in pink, Cas11 (SSU in type I-C and Cse2 in type I-E) in green and yellow, Cas8 (Cas10d in type I-D, Cse1 in type I-E) in purple, Cas3 in red, crRNA in dark green, TS in blue, NTS in cyan. **A** The structure of type I-A Cascade[12], PDB: 7TR8. **B** The structure of type I-B Cascade, PDB: 8H67. **C** The structure of type I-C Cascade[13], PDB: 7KHA. **D** The structure of type I-D Cascade[17], PDB: 7SBA. **E** The structure of type I-E Cascade[19], PDB: 5U07. **F** The structure of type I-F Cascade[21], PDB: 6B45. **G** The structure of type I-G Cascade[24], PDB: 8ANE.

decreases progressively from subtype I-A to I-G. The large subunit of Cascade shares common functions, including facilitating PAM recognition and subsequent dsDNA binding. Additionally, it provides the surface position for the binding of nuclease Cas3[11], despite the absence or low sequence similarity between them. While the involvement of large subunits in PAM recognition is common and well-recognized in type I systems, the participation of Cas5, originally responsible for binding and stabilizing the 5' handle of crRNA, in PAM recognition is rare. This involvement has only been observed in the type I-D[17] and type I-B systems. In type I-C[13] and I-G[24] systems, Cas5 or Csb2 can functionally replace Cas6 in crRNA processing and maturation. This characteristic contributes to the streamlined nature of the I-C and I-G systems.

Apart from the structural and functional differences of Cascade in type I system, there are two different mechanisms for Cas3 recruitment. Type I-B (Supplementary Fig. S7), I-C, I-D, I-E, and I-F exhibits the canonical trans-recruitment mechanism, in which Cas3 recruitment depends on target DNA binding and full R-loop formation[13–23]. Type I-A and possibly I-G systems, however, use another allosteric-activation mechanism to degrade the substrate[12,24]. In type I-A systems, Cascade and Cas3 function as an integral effector complex, and the HD nuclease

domain of Cas3 remains autoinhibited and becomes activated upon full R-loop formation.

Though most subtypes of type I systems, especially type I-C, I-E, and I-F systems, have gained immense attention in recent years, the type I-B CRISPR-Cas system remains poorly understood. We implemented I-B Cascade-Cas3-mediated target genome degradation to characterize the gene editing features in eukaryotes. By inducing DNA lesions in the CD3+ T-cell line at two sites of the *TRAC* locus, we found that *Syn* I-B CRISPR created target-specific 4.5 kb deletions that were all unidirectional to the upstream of the PAM. Though the potential of *Syn* I-B generating long-range deletion hasn't been validated due to the restricted length of the *TRAC* locus, the observed editing features of type I-B are consistent with that of type I-E[31] and *Nla* I-C[32]. Up to 41.2% editing efficiency targeting *TRAC* locus in CD3+ T-cell by *Syn* I-B in our work was obtained. The genome editing efficiency of the *Nla* I-C system using mRNA delivery was poor when the crRNA was designed as a multimeric pre-CRISPR transcript[32]. Therefore, we compared the impact of different crRNA designs on editing efficiency (Supplementary Fig. S8) and posited that the pre-CRISPR RNA maturation may be the rate-limiting step of Cascade assembly. Indeed, using matured crRNA instead of pre-CRISPR significantly boosted the efficiency of this

study. In summary, our work where *Syn* type I-B CRISPR-Cas system was demonstrated as a powerful gene editing tool with exceptionally robust editing efficiency when delivered in mRNA form expands the genomic editing toolbox.

# Methods

## DNA oligonucleotides
All the sequences (HPLC purified) used in this study were shown in Supplementary Table 1 and purchased from Sangon Biotech (Shanghai) Co., Ltd (Shanghai, China).

## Plasmids construction
The genes *cas6b* and *cas11b* were inserted in order into a polycistronic pRSFDuet-1 vector using the BamH I-Hind III and BamH I-Xho I restriction endonuclease cut sites respectively, while the genes *cas8b*, *cas7b* and *cas5b* were cloned into the pCDFDuet-1 vector at BamH I-Xho I restriction sites. To construct the Cas3 expression plasmid, the synthetic cas3 gene was cloned into the pET28a vector using Nde I and Xho I restriction sites. All the constructions were verified by Sanger sequencing (Sangon Biotech). The pUC19-CRISPR array plasmid, commercially synthesized, was co-transformed along with pCDF-*cas8b*-*cas7b*-*cas5b* and pRSF-*cas6b*-*cas11b* into the competent *E. coli* BL21 (DE3) cell for the expression of Cascade complex.

## Protein purification
*E. coli* BL21 (DE3) harboring WT and mutants of Cascade complex (Cas8b, Cas5b, Cas7b, Cas6b, Cas11b, and crRNA) was grown in LB broth supplemented with 50 µg/ml kanamycin, ampicillin, and streptogramin at 37 °C till the $OD_{600}$ reached 0.6. At this point, expression of Cascade complex was induced with 0.5 mM isopropylthio-β-D-galactoside (IPTG), and cells were allowed to grow for 12 h at 25°C. Cells were harvested and resuspended in binding buffer (20 mM Tris-Cl pH 7.5, 500 mM NaCl). Cells were lysed using an ultrasonic cell disruptor, and cell debris was removed by centrifugation at 4 °C and $19,000 \times g$ for 30 min. After centrifugation, the clear supernatant was loaded onto pre-equilibrated 3 ml Strep-Tactin affinity column (IBA Lifesciences). After loading, the column was washed with 10 column volume (CV) of binding buffer to remove unbound proteins and then Cascade complex was eluted in binding buffer containing 5 mM d-Desthiobiotin. The eluate obtained was further purified by size-exclusion chromatography (SEC, Superdex™ 200 Increase 10/300 GL column, Cytiva). Subsequently, Cas proteins and crRNA in the Cascade complex were assayed by SDS-PAGE and Urea-PAGE respectively. After concentration, samples were flash-frozen in liquid nitrogen and stored at −80 °C until use.

*E. coli* BL21 (DE3) harboring Cas3 was grown in LB Broth supplemented with 50 µg/ml kanamycin at 37 °C till the $OD_{600}$ was equal to 0.6. Cas3 expression was induced with 0.5 mM IPTG and cells were allowed to grow overnight at 25°C. Cells were harvested and resuspended in binding buffer (20 mM HEPES pH 7.5, 500 mM NaCl, 20 mM imidazole, 5% (v/v) glycerol). Cell lysate was generated by ultrasonication and then further processed by centrifugation at 4 °C and $19,000 \times g$ for 30 min. After initial fractionation steps, the clarified supernatant was passed through a pre-equilibrated $Ni^{2+}$ affinity column (Cytiva). After washing with 10 CV of binding buffer, Cas3 was eluted at the end of a gradient imidazole elution, where the buffer was composed of 20 mM HEPES (pH 7.5), 500 mM NaCl, 50-500 mM imidazole, 5% (v/v) glycerol. Samples were pooled up and further purified by SEC (Superdex™ 200 Increase 10/300 GL column, Cytiva). Concentrated samples were flash-frozen in liquid nitrogen and stored at −80 °C until further use.

## PAM library generation
To generate a PAM library, a set of complementary oligonucleotides PF-Mix-PAM and PR-Mix-PAM were commercially synthesized (Sangon Biotech). In each single-strand oligonucleotide, a 3 nt PAM sequence (NNN) was linked to the 5' end of the protospacer sequence, with BamH I and Xho I cut sites flanking the whole sequence. After 5′-OH phosphorylation and annealing, these oligonucleotides were ligated into pET-28a vector with T4 Fast ligase and then the ligation products were transformed into DH5α. The generated pET28a-NNN-protospacer plasmid library encompassed 64 potential PAM sequences, the coverage of which was ascertained through Sanger sequencing (Sangon Biotech).

## PAM determination
To ascertain the PAM preferences of the I-B system, the binding affinity between the Cascade complex and all potential PAM sequences was evaluated. Using primers PF-161 and PR-161, segments of 161 bp dsDNA were amplified via PCR. A subsequent amplification was carried out using primers PF-6-FAM and PR-161 to produce 5′-FAM labeled target DNA. The coverage of all 64 PAM sequences was verified by Sanger sequencing to ensure that the DNAs in the library were relatively homogeneous prior to subsequent assays (Supplementary Fig. 9A). The 161 bp 6-FAM-PAM library DNA molecules (320 nM) were then incubated with Cascade complex (0, 10, 20, 40, 100, 200, 400, 800, 2000 nM) in buffer containing 20 mM HEPES (pH 7.5) and 100 mM NaCl at 25 °C for one hour. Incubated samples were electrophoresed on 2% agarose gel and then visualized in Gel Imager. Bands that exhibited specific binding to low concentrations of the Cascade complex were selected, and the extracted DNA from these bands underwent PCR amplification using the primers PF-161 and PR-161, respectively. The PCR products were then analyzed via Sanger sequencing to determine the PAM preference (Supplementary Fig. 9B).

To determine the PAM preferences at −2 and −1 positions, 16 single-stranded DNAs (59 nt) with A at PAM-3 were synthesized (Sangon Biotech). The oligonucleotides were annealed and subsequently amplified through two rounds of PCR using the primer pairs PF-97/PR-97 and PF-97/PR-CY5, yielding 5′-CY5 labeled 97 bp ANN-PAM DNAs. Each PCR product containing a certain PAM sequence underwent incubation with increasing concentrations of the Cascade complex (0 to 200 nM) at 25°C for one hour, after which the binding affinity was determined through an electrophoretic mobility shift assay. The oligonucleotides and primer sequences for this study are listed in Supplementary Tables 1 and 2.

## Electrophoretic mobility shift assay
A final concentration of 10 nM fluorescently labeled target DNA was incubated with titrations of Cascade complex in a 20 µL total reaction volume containing 20 mM HEPES pH 7.5, and 100 mM NaCl. After one hour incubation at 25 °C, 10 µL of each sample was loaded onto 6% acrylamide gel. Electrophoresis was performed in 0.5× TBE buffer at 150 V for 35 min in cold room. DNA was visualized by fluorescence imaging in Tanon MINI Space 3000 system and images were quantified using ImageJ software. The fraction of DNA bound (amount of bound DNA divided by the sum of free and bound DNA) was plotted versus the concentration of Cascade and fit to one site-specific binding with Hill slope using GraphPad Prism 8.0.1. Each PAM sequence was tested in at least three independent experiments.

## In vitro assembly of Cascade-DNA complex
The oligonucleotides synPAM-14F and synPAM-14R that contain the PAM sequence as "ATG" and the protospacer were annealed to generate a double-stranded DNA. The dsDNA was incubated with Cascade at 25 °C for one hour at a molar ratio of 3:1. The sample was centrifugated and loaded onto Superdex™ 200 Increase 10/300 GL column (Cytiva) to remove excess DNA molecules. SDS-PAGE analysis was also conducted to further confirm the assembly of the Cascade-DNA complex in vitro.

## Cryo-EM data acquisition

Three microliters of 1 mg/ml SEC-purified Cascade-DNA complexes were applied to a gold grid which had been glow discharged for 45 seconds. After being stained with phosphotungstic acid hydrate, samples were loaded into field emission transmission electron microscope (Thermo Fisher) for morphological observation. Samples were then concentrated to 4.6 mg/ml and applied to a gold grid (1.2/1.3 300 mesh), which had been glow discharged for 45 s. The grids were blotted at 16 °C, 100% humidity, and plunge-frozen in liquid ethane using the Vitrobot Mark IV (blot time 5 seconds, wait time 30 s). Cryo-EM images were manually collected on a FEI Titan Krios G3i electron microscope equipped with a K2 Summit electron detector (Gatan) which was operated at 300 kV. Images were collected in counting mode, with a nominal defocus range of −1.0 to −2.1 μm at a nominal magnification of 130,000×, corresponding to a calibrated pixel size of 1.1 Å/pixel. The total exposure time of each movie stack was 7.5 s, leading to a total accumulated dose of 50 e-/Å$^2$, which fractionated into 40 frames. The data collection parameters are listed in Supplementary Table 4.

## Cryo-EM data processing and model building

Motion correction, CTF (contrast transfer function) estimation, particle picking, 2D classification, 3D classification, and non-uniform 3D refinement were performed in CryoSPARC (version 4.2). A series of standard refinement procedures including 2D and 3D classification were performed to obtain the final maps as shown in Fig. S2. The initial models of each Cas protein were generated using Alpha fold[53]. The initial models were first docked into the cryo-EM density map in UCSF Chimera[54] and manually rebuilt using Coot[55]. Models were subsequently adjusted in Coot[55] and refined using phenix.real_space_refine[56]. The quality of the structural model was checked using the MolProbity program in Phenix[57]. The detailed refinement statistics are listed in Supplementary Table 4.

## Primary T-cell culture

Healthy human peripheral blood mononuclear cells (PBMCs) were purchased from StemCell Technologies (Cat# 70025) and used according to the manufacturer's instructions. CD3$^+$ T cells were then further isolated by magnetic negative selection using an EasySep Human T-Cell Isolation Kit (STEMCELL, Cat# 17951). Immediately after isolation, T cells were cultured in Gibco CTS AIM V Medium (Thermo Fisher) and stimulated for 2 days with anti-human CD3/CD28 magnetic dynabeads (Thermo Fisher, Cat# A56992) at the beads-to-cells concentration ratio of 1:1 supplemented with human IL-2 at 200 U/ml (Peprotech). After electroporation, T cells were cultured in media with IL-2 at 100 U/ml. Throughout the culture period, T cells were maintained at an approximate density of 1 million cells per ml of media. Every 2-3 days after electroporation, additional media was added, along with additional fresh IL-2 to bring the final concentration to 100 U/ml.

## RNA synthesis

Synthetic CRISPR RNA (crRNA) was chemically synthesized (GenScript Biotech), resuspended to 160 μM, aliquoted and stored at −80 °C. The first and last 3 bases of the crRNA were chemically modified with 2′ O-Methyl.

## In vitro transcription of mRNAs

The full DNA sequences encoding the six Cas proteins were cloned into an IVT template plasmid carrying a T7 promoter, 5′ and 3′ UTR elements, and a poly(A) tail. Endotoxin-free and linearized plasmid preparation service was provided by GenScript Biotech and used as DNA templates. 5′ capped and 3′ polyadenylated mRNAs were synthesized with mMessage mMachine T7 Ultra kit (Thermo Fisher) using m1Ψ−5′-triphosphate (TriLink N-1081) instead of UTP and contained 120 nucleotide-long poly(A) tails. All mRNAs were purified by cellulose purification and analyzed by agarose gel electrophoresis, then stored at −20 °C.

## mRNA electroporation

mRNA and crRNA were electroporated 48 h after initial T-cell stimulation, de-beaded cells were centrifuged for 10 min at 90 g, aspirated, and resuspended in the Lonza electroporation buffer P3 using 20 μL buffer per 1 million cells. For optimal editing, T cells were electroporated per well using a Lonza 4D electroporation system with pulse code EH115. Unless otherwise indicated, 2 μL mRNAs (50, 120, 120, 120, 140, 120 ng of *cas3*, *cas5b*, *cas6b*, *cas7b*, *cas8b*, *cas11b*) were electroporated, along with 2 μg crRNA. Immediately after electroporation, 80 μL of pre-warmed media was added to each well, and cells were allowed to rest for 10 min at 37 °C in a cell culture incubator while remaining in the electroporation cuvettes. After 10 min, cells were moved to 24-well tissue culture plates.

## Flow cytometry

TCR surface disruption was quantified using flow cytometry analysis 5 days post-electroporation. Transfected primary T cells were individualized and analyzed on an Accuri C6 Plus or LSR Fortessa (BD). Surface staining for flow cytometry was performed by pelleting cells and resuspending in 25 μl of PBS with 2% FBS and APC anti-human TCR α/β (Biolegend, Cat# 306717) for 20 min at 4 °C in the dark. As isotype controls were used APC human IgG1 Isotype control recombinant antibody (Biolegend, Cat # 403505). Cells were washed twice in FACS buffer before resuspension. Flow cytometry data were analyzed with FlowJo v10.7.1.

## DNA lesion analysis by long-range PCR and PacBio sequencing

The genomic DNA of the edited cells was isolated using a Puregene Cell Kit (8 × 10$^8$) (Qiagen) per the manufacturer's instructions. Long-range PCR reactions were carried out with a KOD FX Neo kit (TOYOBO), each PCR used gDNA as a template and the complement primers were detailed in Supplementary Tables. PCR products (9.4 kb) were resolved on 0.8% agarose gel, stained with SYBR Safe (Invitrogen), and visualized using a ChemiDoc MP imager (BioRad). To precisely define Cas3-induced deletions or insertions, the PCR products were analyzed by PacBio sequencing.

## Reporting summary

Further information on research design is available in the Nature Portfolio Reporting Summary linked to this article.

# Data availability

The cryo-EM reconstructed density map of *Syn* Cascade-dsDNA complex in partial and full R-loop formations have been deposited into the Electron Microscopy data bank under accession numbers EMD-34495 and EMD-35629, respectively. The associated atomic coordinate has been deposited into the Protein Data Bank with PDB 8H67 and PDB 8IP0. All materials and data are available upon request from the corresponding authors Meiling Lu (lumeiling@cpu.edu.cn), Zhenhuang Yang (yanginchina@hotmail.com), and Yibei Xiao (yibei.xiao@cpu.edu.cn). Source data are provided in this paper.

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

## Acknowledgements

This work is supported by the National Key Research and Development Program of China (2018YFA0902000) (to Y.B.X. and M.L.L.), and the National Natural Science Foundation of China (31970547) (to Y.B.X.). We would like to thank you for cryo-EM data collection at the Instrument Analysis Center (IAC) at Shanghai Jiao Tong University.

## Author contributions

M.L.L., Z.H.Y. and Y.B.X. conceived the project and designed the experiments. M.L.L., C.L.Y., Y.W.Z., W.J.J., Z.Y., C.Y.H., J.Z.M. and Z.H.Y. carried out the experiments. M.L.L., C.L.Y., Y.W.Z., Z.H.Y. and Y.B.X. analyzed the data. M.L.L., C.L.Y., Z.H.Y. and Y.B.X. wrote the manuscript. All authors discussed the results and contributed to the final manuscript.

## Competing interests

The authors declare no competing interests.
