## [Peer Review File · Nature Communications]

Reviewers' Comments:

Reviewer #1:

Remarks to the Author:

The manuscript by Lu et al is a high-quality study. The highlights include 1) a complete and high-resolution mechanism for the RNA-guided DNA targeting process by the type I-B Cascade complex; 2) demonstration of syn I-B CRISPR system in long-range DNA deletion in human cells; and 3) improving the mRNA-based targeting efficiency to the best reported values. This study is of general interest to readers in both the mechanistic field and the gene editing field. This reviewer only has a minor points for the reviewers to further polish the manuscript.

1. Nomenclature is confusing for this system – Cas8 in this system is named Cas7 in other type I systems, and the Cmx8 is the Cas8 subunit in other type I systems. This would create a lot of confusion down the road. I suggest the authors label both sets of names in figure 1A and mention in the text that they will follow the naming convention from then on, for the rest of the text and figures.

2. Two sets of PAM definition were used – PAM residues were first mentioned as first residue, second residue, third residue, then referred to as PAM-1, PAM-2, and PAM-3. I suggest the authors stick with PAM-1, -2, -3, etc, and explain in the text once that this refers to their position relative to the R-loop residues. Make sure to correct both the text and the figures.

3. PAM code characterization – It is not clear to me how statistically sound the PAM code is in Figure 1E. The code is rather promiscuous, and it is hard to claim that Syn I-B prefers G at PAM-1, because A seems to be only slightly disfavored. From reading the methods, it is clear whether the authors did deep-sequencing or sanger-sequencing to define the PAM code. How many reads were used to derive the code here?

4. Page 8, line 216: Please add a supplemental figure showing the EM density for the TS/NTS/crRNA in the partial R-loop and full R-loop states.

5. Page 8, line 218-221: The rendering in Figure 3E is different to follow. The conformational changes are too busy, and the labels are too small.

6. Page 11, line 308-09: The authors claimed that using the mature crRNA led to more efficient genome editing as compared to using the pre-crRNA transcript. This claim did not refer to any data/figure. The control for using the pre-crRNA transcript should be shown.

7. Figure 5: Label I-A, I-B, I-C, and I-D on the panels, this reduces a lot of back and forth referencing.

Reviewer #2:

Remarks to the Author:

Structure and genome editing of type I-B CRISPR-Cas
Lu and Yu et al.

Lu and Yu et al. present a study that determines the PAM requirements of CRISPR-Cas type I-B Cascade, structure of the complex bound to DNA target, and demonstrate editing in mammalian cells. Importantly, this is the first report of the type I-B Cascade, the missing structure of type I systems. The authors observe several interesting structural differences compared to other type I systems, e.g. an arginine from Cas5 subunit makes key interactions with DNA in determining the PAM preference. The authors also demonstrate gene deletions in mammalian cells with efficiency up to 41%. Results from this work make good advancements in our understanding of type I CRISPR-Cas systems. However, the analysis is let down by fundamental misunderstandings about CRISPR-Cas, interpretation of the DNA binding studies and clarity of the structural figures (which made judging the validity of the structural interpretations very difficult). Below I have lists of major and minor recommendations to improve the manuscript.

Major:

1. The authors have misnamed the Cas7 subunit as Cas8. This mistake is throughout the document (main text, figures, supp, structural reports). This is essential to correct. Also, It should be noted in the text that the Cmx8 subunit has been renamed Cas8b (Makarova et al 2020).

2. Line 121 The author states "Comparison of affinities...". Affinity was not quantified and therefore difficult to compare different PAMs, especially those with similar binding ability. This is an issue throughout this part of the work, resulting in over or misinterpretation. For example, the A-Y-G motif appears not as strict. A Guanine in the first position appears well tolerated (Sup. Fig 1B A-T-G vs G-T-G), as does an Adenosine in the third position looks well tolerated too (Sup. Fig 1A A-C-G vs A-C-A). These differences will also effect discussion in relation to the structure in the results and discussion sections. The authors should quantify key PAM motifs and adjust their comparisons to the structural work and conclusions accordingly.

3. The structural figures in Figure 3 are of poor quality, including too small text. Figure 3A is particularly poor that base interactions can't be seen. This is very important because this is the basis of understanding PAM preference from a structural viewpoint. Structural figures and text need to be clearer.

4. How was the model built? The methods are insufficient and require details. Figure 2F shows an alignment of modelled and predicted structures, but the main text states "The final 3D reconstruction reached an overall resolution higher than 3.6 Å and enabled de novo model building." Which was it? Figure 2F alignment is not discussed in the text and it is not apparent what it adds to the manuscript.

5. Is Cas5 N102 conserved? What about other key features involved in PAM determination?

Minor:

General comments. Overall written well. Look for odd typo and use of the wrong tense. Referencing other people's work is too light in some sections.

Line 106 Why was it expected that Cas11 would be expressed and that the cmx8 gene has an internal RBS (Line 100)? The primary publication for this was McBride et al (2020) and they showed this phenomenon occurred in type I-D, I-C and I-B CRISPR-Cas systems. The text should be rectified to include this.

There is inconsistent naming of the system and Cascade complex. Convention is the system be called type I-B system (not Syn I-B system). There is no convention for the name of the Cascade complex, but the authors should be consistent. Line 82 and 83 the authors call the complex type I-B SynCascade and Syn I-B Cascade. Generally these would be called I-B Cascade.

Line 62: The primary reference for each structure of the other type I systems should be cited.

Line 67: Lack of Cas3 recruitment to the type I-D Cascade was shown by Lin et al. 2020 and McBride et al. 2020.

Line 107 How was the size of crRNA determined? The gel on Fig 1D does not provide sufficient information to conclude 71 nucleotides.

Line 114 There would be benefit to the reader to provide some further information about the "affinity assay". This would provide context to the next sentence about non-specific binding, which currently is confusing.

Line 142 Some further clarification between full R-loop and partial R-loop would be useful. A full R-loop is not actually observed (i.e. not full NTS).

Line 147 References for I-A and I-C structures are required. Also McBride et al. and Schwartz et al 2022 should be included for the I-D Cascade structure, which first showed a longer helical backbone compared to I-E and I-F.

Line 151 Jore et al 2011 showed the 6th base flips out, a feature for all class 1 complexes.

Line 153 It is not surprising that low sequence similarity was observed between Cmx8 and other large subunits (note, I doubt there was "no sequence similarity") as CRISPR can have very great

sequence diversity. How does Cmx8 structure compare to other large subunits?

Line 167 McBride et al 2020 first compared Cas11 protein sequences across different CRISPR-Cas types and showed low sequence identity. Remove no detectable sequence identity.

Line 173 The best studied PAM recognition by type I systems involves three components, glycine loop, Gln-wedge and Lys-finger (see Hynes et al 2016 for type I-E system, and Schwartz et al 2022 and McBride et al. 2023 for type I-D system; it also appears I-A may use this via K137 in Hu et al. 2022). Some systems do not have the finger, instead type I-F (Rollins et al 2019) and type I-C (O'Brien et al 2023) have and Asn that makes specific interactions with the PAM, usually a pyrimidine. Text should be modified to reflect this.

Line 181 and Line 188 What residues are GNS loop in Cas5? To the point above, it appears the N has a similar role to the I-F and I-C systems. It is interesting that in the I-B system the N comes from Cas5 and not Cas8. A similar "subunit swap" occurs in the I-D system, where the wedge is from Cas5 instead of Cas8 (Discussed in McBride et al 2023).

Line 184 Which Carbonyl O does N102 interact with? There are two Carbonyl O in Thymine. Is it the carbonyl O group that is in an equivalent position in Cytosine? The author states the interaction might explain why an Adenosine is strongly preferred in the first PAM position. As raised above, there appears tolerance (or perhaps equal affinity) for a Guanine in the first position. Could N102 be able to bind Cytosine similarly to how it binds Thymine?

Line 192 Suggest changing the start of the sentence to "Residue P156 of the "GVP" motif in Cxm8..."

Line 197 Show figure displaying the interactions. Why then is Adenosine also tolerable in this position?

Line 233 Cite O'Brien et al 2023.

Line 250 Some explanation is required here and in the methods on how TCR is visualised by flow cytometry.

Line 254 Change to flow cytometry

Line 259 Why were the <1 kb deletions not shown in Fig 4E? These were comprised the majority of observed deletions.

Line 284 Cite relevant literature

Line 308 Discussed "using matured crRNA instead of pre-CRISPR significantly boosted the efficiency in this study". I don't believe the direct comparison was done in this study as only matured crRNA was used. The authors need to do the direct comparison to make this conclusion, otherwise alter phrasing.

L442 Suggest rephrasing the start of this sentence "The full DNA sequences encoding the six Cas proteins..."

Figure 1: A) Change to Cas7. B) How was the complex size of 450 kDa determined?

Figure 3: What do the blue and black dashes represent? Explain NTS and colours. Label subunits.

Sup Figure 3: A description is required, only results are mentioned. What do the symbols represent? E.g. the T?

Sup Figure 4: A description is required, only results are mentioned.

Sup Figure 5: Suggest showing the N and O groups involved in H-bonds.

Point-by-point Response to Referees

Response to Reviewer #1 :

The manuscript by Lu et al is a high-quality study. The highlights include 1) a complete and high-resolution mechanism for the RNA-guided DNA targeting process by the type I-B Cascade complex; 2) demonstration of syn I-B CRISPR system in long-range DNA deletion in human cells; and 3) improving the mRNA-based targeting efficiency to the best reported values. This study is of general interest to readers in both the mechanistic field and the gene editing field. This reviewer only has a minor points for the reviewers to further polish the manuscript.

Response: Thank you for the positive evaluation and helpful suggestions. We now have polished the manuscript accordingly.

1. Nomenclature is confusing for this system – Cas8 in this system is named Cas7 in other type I systems, and the Cmx8 is the Cas8 subunit in other type I systems. This would create a lot of confusion down the road. I suggest the authors label both sets of names in figure 1A and mention in the text that they will follow the naming convention from then on, for the rest of the text and figures.

Response: Thank you for pointing our mistake. We have now revised the naming conventions to accurately refer to the large subunit as Cas8b and the backbone protein as Cas7b across all sections and figures in the text. An example of the revision for the Cas proteins' nomenclature in Figure 1A is shown as below.

Figure 1. Heterologous expression and PAM sequence determination of type I-B Cascade

2. Two sets of PAM definition were used – PAM residues were first mentioned as first residue, second residue, third residue, then referred to as PAM-1, PAM-2, and PAM-3. I suggest the authors stick with PAM-1, -2, -3, etc, and explain in the text once that this refers to their position relative to the R-loop residues. Make sure to correct both the text and the figures.

Response: Thank you for this suggestion. We have standardized the definition of the PAM sequence in terms of PAM-1, PAM-2 and PAM-3 throughout the text (Line 119-123, Line 130, Line 193, Line 196, Line 203, Line 207-208) and in corresponding figures (Figure 3B-3E).

Figure 3. Structural basis for PAM recognition and NTS stabilization

3. PAM code characterization – It is not clear to me how statistically sound the PAM code is in Figure 1E. The code is rather promiscuous, and it is hard to claim that Syn I-B prefers G at PAM-1, because A seems to be only slightly disfavored. From reading the methods, it is clear whether the authors did deep-sequencing or sanger-sequencing to define the PAM code. How many reads were used to derive the code here?

Response: For the identification of PAM, DNA captured by the Cascade in each of the selected lanes (lanes 3 to 6) was carefully excised for further amplification and subsequent Sanger sequencing (added in Figure 1E and shown as below).

Figure 1. Heterologous expression and PAM sequence determination of type I-B Cascade

The DNA sequencing outcomes from these four lanes were consistent, and the PAM preferences were depicted as a computational logo in Figure 1E. We quantified the EMSA results as shown in

Supplementary Figure 1B. The analysis indicated that guanine (G) was the most favored nucleotide at the PAM-1 position, with adenine (A) being slightly less preferred compared to G. This finding aligned with the Sanger sequencing results.

4. Page 8, line 216: Please add a supplemental figure showing the EM density for the TS/NTS/crRNA in the partial R-loop and full R-loop states.

Response: We have incorporated the EM density for the nucleic acids in both the full R-loop (Line 217) and partial R-loop states (Line 227), as illustrated in Figure 2D and 2E.

Figure 2. Cryo-EM snapshots of the type I-B Cascade-dsDNA complex

5. Page 8, line 218-221: The rendering in Figure 3E is different to follow. The conformational changes are too busy, and the labels are too small.

Response: We have restructured the original Figure 3E into a new set of figures, now presented as Figure 4A-4E, and have enlarged the labels for clearer visibility.

Figure 4. Conformational changes during full R-loop formation

6. Page 11, line 308-09: The authors claimed that using the mature crRNA led to more efficient genome editing as compared to using the pre-crRNA transcript. This claim did not refer to any data/figure. The control for using the pre-crRNA transcript should be shown.

Response: We have added a comparison of genome editing efficiency using mature crRNA versus

the pre-crRNA transcript as shown below (Supplementary Figure 8).

Supplementary Figure 8. The comparison of genome editing efficiency using crRNA of different type

7. Figure 5: Label I-A, I-B, I-C, and I-D on the panels, this reduces a lot of back and forth referencing.

Response: We have added labels on the panels and the citations in the figure legend.

Figure 6. Comparison of type I CRISPR effector structures

Reviewer #2 (Remarks to the Author):

Structure and genome editing of type I-B CRISPR-Cas

Lu and Yu et al.

Lu and Yu et al. present a study that determines the PAM requirements of CRISPR-Cas type I-B Cascade, structure of the complex bound to DNA target, and demonstrate editing in mammalian cells. Importantly, this is the first report of the type I-B Cascade, the missing structure of type I systems. The authors observe several interesting structural differences compared to other type I systems, e.g. an arginine from Cas5 subunit makes key interactions with DNA in determining the PAM preference. The authors also demonstrate gene deletions in mammalian cells with efficiency up to 41%. Results from this work make good advancements in our understanding of type I CRISPR-Cas systems. However, the analysis is let down by fundamental misunderstandings about CRISPR-Cas, interpretation of the DNA binding studies and clarity of the structural figures (which made judging the validity of the structural interpretations very difficult). Below I have lists of major and minor recommendations to improve the manuscript.

Response: Thank you for appreciating the significance of our study and helpful suggestions.

Major:

1. The authors have misnamed the Cas7 subunit as Cas8. This mistake is throughout the document (main text, figures, supp, structural reports). This is essential to correct. Also, It should be noted in the text that the Cmx8 subunit has been renamed Cas8b (Makarova et al 2020).

Response: Thank you for pointing out the incorrect nomenclature of the CRISPR-Cas system components in our manuscript. We have now revised the naming conventions to accurately refer to the large subunit as Cas8b and the backbone protein as Cas7b throughout the document and figures. And the Cmx8 subunit renamed as Cas8b was also noted in the text (Line 98-99).

Below is the revision of the nomenclature in Figure 1A.

Figure 1. Heterologous expression and PAM sequence determination of type I-B Cascade

2. Line 121 The author states “Comparison of affinities...”. Affinity was not quantified and therefore difficult to compare different PAMs, especially those with similar binding ability. This is an issue throughout this part of the work, resulting in over or misinterpretation. For example, the A-Y-G motif appears not as strict. A Guanine in the first position appears well tolerated (Sup. Fig 1B A-T-G vs G-T-G), as does an Adenosine in the third position looks well tolerated too (Sup. Fig 1A A-C-G vs A-C-A). These differences will also effect discussion in relation to the structure in the results and discussion sections. The authors should quantify key PAM motifs and adjust their comparisons to the structural work and conclusions accordingly.

Response: Much appreciation for your suggestion. We quantified EMSA results (5'-AYN) to compare the preference of the PAM with similar binding ability. The analysis indicated that 5'-AYG

exactly was the most favorable PAM sequence, while A at PAM-1 was slightly less preferred than G (Supplementary Figure 1B).

Supplementary Figure 1. PAM identification by EMSA

When we constrained PAM-2 and PAM-3 to Y and R, respectively, to re-evaluate the preference for PAM-1, we observed a deviation from previous result. Notably, the Cascade complex exhibited a reduced affinity for the target DNA when PAM-3 was not A (Supplementary Figure 1C and 1D).

Supplementary Figure 1. PAM identification by EMSA

Following your suggestion, we further examined the binding affinity between the Cascade and target DNA containing 5'-GYR PAM. This investigation (shown as below) revealed a significantly reduced binding affinity.

(Not included in the manuscript)

The explanation for this preference might be related to the interaction between N102 of Cas5b and the nucleotide at the PAM-3 position, where a hydrogen bond can be formed when it was A, but not when it was G.

Figure 3. Structural basis for PAM recognition and NTS stabilization

3. The structural figures in Figure 3 are of poor quality, including too small text. Figure 3A is particularly poor that base interactions can't be seen. This is very important because this is the basis of understanding PAM preference from a structural viewpoint. Structural figures and text need to be clearer.

Response: The original Figure 3 has been reorganized into two separate figures, now presented as Figure 3 and Figure 4, to enhance clarity. Additionally, the labels within these figures have been enlarged to ensure they are easily readable.

Figure 3. Structural basis for PAM recognition and NTS stabilization

Figure 4. Conformational changes during full R-loop formation

4. How was the model built? The methods are insufficient and require details. Figure 2F shows an alignment of modelled and predicted structures, but the main text states “The final 3D reconstruction reached an overall resolution higher than 3.6 Å and enabled de novo model building.” Which was it? Figure 2F alignment is not discussed in the text and it is not apparent what it adds to the manuscript.

Response: Thank you for pointing out our mistake. In this study, templates predicted by AlphaFold were utilized for model building. We have added the information regarding the model building in the main text “The final 3D reconstruction reached an overall resolution higher than 3.6 Å, which was sufficient to identify the direction of the main chain and the clear side chains” (Line 140-142) and in the Methods section “The initial models of each Cas protein were generated using Alpha fold⁵³. The initial models were first docked into the cryo-EM density map in UCSF Chimera⁵⁴ and manually rebuilt using Coot⁵⁵. Models were subsequently adjusted in Coot⁵⁵ and refined using phenix. real_space_refine⁵⁶. The quality of the structural model was checked using the MolProbity program in Phenix⁵⁷. The detailed refinement statistics are listed in Supplementary Table 4” (Line 436-442). The original Figure 2F was moved to the Supplementary information as Supplementary Figure 4.

5. Is Cas5 N102 conserved? What about other key features involved in PAM determination?

Response: Cas5 N102 is not conserved, as evidenced by the alignment of Cas5b from different Type I-B CRISPR-Cas systems (N102 was indicated using blue arrow).

(Not included in the manuscript)

While both N332 (involved in PAM-1-recognition) and P156 (involved in PAM-2 recognition) of Cas8b are conserved (indicated using blue arrow).

Supplementary Figure 3. Multiple sequence alignment of *Syn*Cas8b and other large subunits of type I-B CRISPR system

Minor:

General comments. Overall written well. Look for odd typo and use of the wrong tense. Referencing other people's work is too light in some sections.

Response: Thank you for all the suggestions to improve the quality of our manuscript.

Line 106 Why was it expected that Cas11 would be expressed and that the *cmx8* gene has an internal RBS (Line 100)? The primary publication for this was McBride et al (2020) and they showed this phenomenon occurred in type I-D, I-C and I-B CRISPR-Cas systems. The text should be rectified to include this.

Response: We have changed the sentence in the text according to your comments "*Like its counterparts in I-C¹³⁻¹⁵ and I-D^{16,17} systems, the cas8b large subunit of this system also includes an internal ribosome-binding site at its 3' terminus, which encodes a separate small subunit Cas11¹⁷*" (Line 99-101).

There is inconsistent naming of the system and Cascade complex. Convention is the system be called type I-B system (not Syn I-B system). There is no convention for the name of the Cascade complex, but the authors should be consistent. Line 82 and 83 the authors call the complex type I-B SynCascade and Syn I-B Cascade. Generally these would be called type I-B Cascade.

Response: We have standardized the naming conventions in our study as "*Syn type I-B Cascade*".

Line 62: The primary reference for each structure of the other type I systems should be cited.

Response: We have added the citations of the other type I systems in this sentence "*All type I-A¹², I-C¹³⁻¹⁵, I-D^{16,17}, I-E¹⁸⁻²⁰, I-F²¹⁻³², I-G²⁴ Cascade structures have been determined to date, except for type I-B Cascade*" (Line 62-63).

Line 67: Lack of Cas3 recruitment to the type I-D Cascade was shown by Lin et al. 2020 and McBride et al. 2020.

Response: We have added the reference about Cas3 recruitment of type I-D Cascade systems in this sentence "*In type I-C¹³⁻¹⁵, I-D^{16,17,26}, I-E¹⁸⁻²⁰ and I-F²¹⁻²³ systems, Cas3 recruitment is dependent on R-loop formation upon target DNA recognition by Cascade*" (Line 67-69).

Line 107 How was the size of crRNA determined? The gel on Fig 1D does not provide sufficient information to conclude 71 nucleotides.

Response: Given the crRNA processing characteristics in Type I systems, where the length of a mature crRNA comprises the length of a spacer plus a repeat, we deduced the length of the co-purified crRNA in the Type I-B system to be 71 nucleotides (nt), based on the lengths of the spacer and repeat. The size of the crRNA was further determined through sequencing analysis on a urea gel. We have introduced three RNA markers of varying lengths to facilitate the comparison of crRNA lengths (Figure 1D).

Figure 1. Heterologous expression and PAM sequence determination of type I-B Cascade

Line 114 There would be benefit to the reader to provide some further information about the “affinity assay”. This would provide context to the next sentence about non-specific binding, which currently is confusing.

Response: The relevant sections have been updated to clarify that the affinity assay was conducted using EMSA “All NNN-protospacers were then amplified into 161 bp 6-FAM-labelled dsDNA and incubated with the SynCascade for Electrophoretic Mobility Shift Assay (EMSA). Specific bands, indicative of DNA binding with low SynCascade complex concentrations, were singled out and followed with Sanger sequencing to map the PAM preference” (Line 116-118).

Line 142 Some further clarification between full R-loop and partial R-loop would be useful. A full R-loop is not actually observed (i.e. not full NTS).

Response: The relevant sentences and figures (Figure 2D, 2E) have been updated to clarify the observed full R-loop and partial R-loop formed states.

The sentence added in Line 145-146 was “However, the non-target strand (NTS) was not fully visible, particularly the bulge for Cas3 recruitment”.

The sentence added in Line 147-149 was “We also observed a subset of particles within our cryo-EM dataset that formed a partial R-loop state, with only 5 nt of the TS hybridized to the crRNA, alongside duplex DNA bound with Cas8b”.

Figure 2. Cryo-EM snapshots of the type I-B Cascade-dsDNA complex

Line 147 References for I-A and I-C structures are required. Also McBride et al. and Schwartz et al 2022 should be included for the I-D Cascade structure, which first showed a longer helical backbone compared to I-E and I-F.

Response: The citations were added correspondingly in the sentence “*It shared structural similarity with that of I-A¹², I-C¹³⁻¹⁵ and I-D^{16,17} Cascades, featuring a longer helical backbone*” (Line 153-154).

Line 151 Jore et al 2011 showed the 6th base flips out, a feature for all class 1 complexes.

Response: This literature has been incorporated into the text “*Each Cas7b subunit occupied 6 nucleotides of the crRNA with a recurring periodic pattern of 5+1 nt, where the sixth base flipped out in the opposite direction to the other five⁴⁴ (Fig. 2G)*” (Line 157).

Line 153 It is not surprising that low sequence similarity was observed between Cmx8 and other large subunits (note, I doubt there was “no sequence similarity”) as CRISPR can have very great sequence diversity. How does Cmx8 structure compare to other large subunits?

Response: We conducted multiple sequence alignments using the Omega cluster. The results, as indicated below, demonstrate that Cas8b possesses very low sequence similarity with the large subunit of other Type I systems. Therefore, we have revised the phrase from “*no sequence similarity*” to “*very low sequence similarity*” to ensure greater accuracy and precision in our description (Line 160).

Percent Identity Matrix - created by Clustal2.1

5U0A_2 -E	100.00	17.55	14.98	13.28	17.34	8.74	10.71
7SBA_3 -D	17.55	100.00	19.44	15.54	16.06	10.99	16.78
6B45_1 -F	14.98	19.44	100.00	12.23	17.08	8.22	8.95
8DFO_3 -C	13.28	15.54	12.23	100.00	20.99	15.32	16.39
8B2X_1 -G	17.34	16.06	17.08	20.99	100.00	16.45	17.99
7TR6_1 -A	8.74	10.99	8.22	15.32	16.45	100.00	22.74
I-B	10.71	16.78	8.95	16.39	17.99	22.74	100.00
	5U0A_2 -E	7SBA_3 -D	6B45_1 -F	8DFO_3 -C	8B2X_1 -G	7TR6_1 -A	I-B

(Not included in the manuscript)

The alignments of Cas8b with the large subunit of other Type I systems, as detailed below, also reveal that they share low structural similarity (Cas8b colored in yellow, and the other large subunits colored in grey).

(Not included in the manuscript)

Line 167 McBride et al 2020 first compared Cas11 protein sequences across different CRISPR-Cas types and showed low sequence identity. Remove no detectable sequence identity.

Response: Corrected correspondingly *“The C-terminal portion, identical to the Cas11b small subunit, is similar in size and secondary structure of the α -helical bundle observed in the small Cas11 subunit in most Class I effectors, though it exhibits low identity with other Cas11 proteins¹⁶”*. (Line 175)

Line 173 The best studied PAM recognition by type I systems involves three components, glycine loop, Gln-wedge and Lys-finger (see Hynes et al 2016 for type I-E system, and Schwartz et al 2022 and McBride et al. 2023 for type I-D system; it also appears I-A may use this via K137 in Hu et al. 2022). Some systems do not have the finger, instead type I-F (Rollins et al 2019) and type I-C (O’Brien et al 2023) have and Asn that makes specific interactions with the PAM, usually a pyrimidine. Text should be modified to reflect this.

Response: We have revised the paragraph as your suggestion *“In type I systems, the best studied PAM recognition typically involves large subunit-mediated DNA minor groove contacts. This recognition involves three components: the specific residues on a Gly-rich loop in the large subunit's NTD that interacts with the DNA's minor groove, a Gln-wedge that inserts itself into the dsDNA path beneath the PAM, and a Lys-finger that favorably forms electrostatic interactions with a*

pyrimidine in the PAM⁴⁵. However, the Gln-wedge may change to an Asn-wedge in type I-C¹⁴ or to a Lys-wedge in type I-F²³, and the Lys-finger is replaced by Asn in type I-C¹⁴ and type I-F²³.” (Line 180-187)

Line 181 and Line 188 What residues are GNS loop in Cas5? To the point above, it appears the N has a similar role to the I-F and I-C systems. It is interesting that in the I-B system the N comes from Cas5 and not Cas8. A similar “subunit swap” occurs in the I-D system, where the wedge is from Cas5 instead of Cas8 (Discussed in McBride et al 2023).

Response: We have specified the residue positions of the GNS loop within the manuscript, as mentioned on Line 190 “*Notably, a “GNS” loop (residues 101-103) of SynCas5b intercalated into the major groove of the PAM, opposite to the “GVP” loop from Cas8b, aiding in PAM recognition (Figs. 3A and 3C)*”.

Line 184 Which Carbonyl O does N102 interact with? There are two Carbonyl O in Thymine. Is it the carbonyl O group that is in an equivalent position in Cytosine? The author states the interaction might explain why an Adenosine is strongly preferred in the first PAM position. As raised above, there appears tolerance (or perhaps equal affinity) for a Guanine in the first position. Could N102 be able to bind Cytosine similarly to how it binds Thymine?

Response: We refined the model of the PAM recognition region. Although the resolution is insufficient to definitively ascertain the orientation of the N102 side chain, the likelihood of hydrogen bond formation between N102 and A_{NT-3} is higher given the proximity. This finding coherently aligns with the preference for A observed at the PAM-3 position. As a result, we have revised the description of the interaction between N102 and nucleotide in both the manuscript text “*Within the “GNS” loop, the N102 residue is closely interact with the amino group of A_{NT-3}, forming a hydrogen bond*” (Line 193) and Figure 3B to incorporate these findings.

Figure 3. Structural basis for PAM recognition and NTS stabilization

Line 192 Suggest changing the start of the sentence to “Residue P156 of the “GVP” motif in Cxm8...”

Response: Corrected correspondingly “*Residue P156 of “GVP” motif in Cas8b is situated close to the minor groove of the PAM sequence*” (Line 202).

Line 197 Show figure displaying the interactions. Why then is Adenosine also tolerable in this position?

Response: N332 is positioned at a distance conducive to forming hydrogen bonds with both G_{NT-1} and C_{T-1}, establishing a dual contact with the two strands. We have updated the description (Line 206-208) “*Within this wedge, N332 establishes two hydrogen bonds (Fig. 3D): one with the N1 of G_{NT-1} and the other with the N3 of C_{T-1}, making PAM-1 more favorable to G*” and included Figure 3D to illustrate the interactions between N332 and the nucleotides.

Figure 3. Structural basis for PAM recognition and NTS stabilization

Line 233 Cite O’Brien et al 2023.

Response: Corrected correspondingly “*These features suggest that the R-loop formation follows a kinetically favorable mechanism, analogous to that observed in type I-C^{14b}*” (Line 243).

Line 250 Some explanation is required here and in the methods on how TCR is visualised by flow cytometry.

Response: We have updated the explanation in both the Result section “*The endogenous TCR- α chain was disrupted by knockout of TRAC gene, and a specific monoclonal antibody was utilized to track TCR- $\alpha\beta$ expression, which only occurs on the T cell surface when both TCR- α and - β chains are co-expressed*” (Line 259-262) and in Methods section “*TCR surface disruption was quantified using flow cytometry analysis 5 days post electroporation. Transfected primary T cells were individualized and analyzed on an Accuri C6 Plus or LSR Fortessa (BD). Surface staining for flow cytometry was performed by pelleting cells and resuspending in 25 μ l of PBS with 2% FBS and APC anti-human TCR α/β (Biolegend) for 20 min at 4 °C in the dark. As isotype controls were used mouse APC-conjugated IgG1 (Biolegend). Cells were washed twice in FACS buffer before resuspension. Flow cytometry data were analyzed with FlowJo v10.7.1.*” (Line 480-487).

Line 254 Change to flow cytometry

Response: Corrected correspondingly (Line 268).

Line 259 Why were the <1 kb deletions not shown in Fig 4E? These were comprised the majority of observed deletions.

Response: Based on the results, deletion fragments smaller than 1 kb constituted 60% of the 73 total samples observed. However, given the unique capability of Type I CRISPR-Cas systems to induce deletion of long fragments—a distinguishing feature from other systems—we have chosen to exclusively present deletion fragments larger than 1kb in the Figure 5E within the main text. The total 73 deletion fragments were also provided in the supplementary information.

Line 284 Cite relevant literature

Response: Corrected correspondingly “*In type I-C¹³ and I-G²⁴ systems, Cas5 or Csb2 can functionally replace Cas6 in crRNA processing and maturation*” (Line 296-297).

Line 308 Discussed “using matured crRNA instead of pre-CRISPR significantly boosted the efficiency in this study”. I don’t believe the direct comparison was done in this study as only matured crRNA was used. The authors need to do the direct comparison to make this conclusion, otherwise alter phrasing.

Response: We have included a comparison of genome editing efficiency using mature crRNA versus the pre-crRNA transcript as Supplementary Figure 8.

Supplementary Figure 8. The comparison of genome editing efficiency using crRNA of different type

L442 Suggest rephrasing the start of this sentence “The full DNA sequences encoding the six Cas proteins...”

Response: Corrected correspondingly “The full DNA sequences encoding the six Cas proteins were cloned into an IVT template plasmid carrying a T7 promoter, 5’ and 3’ UTR elements, and poly(A) tail” (Line 460).

Figure 1: A) Change to Cas7. B) How was the complex size of 450 kDa determined?

Response: We have updated the naming conventions throughout the manuscript to accurately designate the large subunit as Cas8b and the backbone protein as Cas7b in all sections and figures. For the purpose of estimating the size of the Cascade complex, Ferritin (440 kDa) was chosen as the reference protein. We compared the elution volumes to approximate the size of the Cascade complex. Figure 1B has been updated to include the elution curve of Ferritin, enhancing the accuracy of our size estimation. We have also revised the sentence as “Size-exclusion chromatography of the affinity purified sample indicated successful assembly of type I-B Cascade, which was eluted at a volume corresponding to slightly smaller than 440 kDa (Fig. 1B)” in the main text (Line 104-107).

B

Figure 1. Heterologous expression and PAM sequence determination of type I-B Cascade

Figure 3: What do the blue and black dashes represent? Explain NTS and colours. Label subunits.

Response: We have revised the figure legend to clarify the significance of the dashes used and have incorporated labels for the subunits and nucleic acid strands.

“(A) Binding patterns of target DNA with Cas8b and Cas5b. (B) N102 of Cas5b formed a H-bond with A_{NT-3} (blue dash lines). (C) A “GVP” loop (154-156) of Cas8b is proximal to and interacts with the minor groove of the PAM duplex. (D) The wedge N332 of Cas8b forms two H-bonds with G_{NT-1} and C_{T-1} , respectively. The adjacent S333 interacted with the ribose of C_{T-1} . H-bond are depicted using yellow dashed lines. (E) Schematic of the residues involved in PAM recognition for SynCascade. (F, G) Specific residues in Cas8b NTD involved in NTS stabilization. The positively charged residues and aromatic residues form non-specific interactions with the NTS backbone and bases, respectively. H-bond and aromatic forces are illustrated using blue and black dashed lines, respectively. (H) Specific residues in Cas11b small subunits involved in NTS stabilization. H-bond are depicted using blue dashed lines.” (Line 682-692)

Figure 3. Structural basis for PAM recognition and NTS stabilization

Sup Figure 3: A description is required, only results are mentioned. What do the symbols represent? E.g. the T?

Response: Corrected correspondingly “*SynCas8b shares very low sequence homology with the large subunits in Type I systems. Its highest homology with large subunits of the same subtype is approximately 52%, and it has about 30% homology with the large subunit in the Type I-B CAST system. The symbols of a helix, arrow, and letter T denote the secondary structures as α -helix, β -sheet, and β -turn, respectively*” (Supplementary information, Page 4).

Sup Figure 4: A description is required, only results are mentioned.

Response: This figure has been updated to Supplementary Figure 5. The corresponding description of this figure has been added “(A) Sanger sequencing of the captured DNA by G101A Cascade from the PAM library using EMSA. (B) The EMSA of 5'-NTG PAM with G101A Cascade. (C) Sanger sequencing of the captured DNA by N102A Cascade from the PAM library using EMSA. (D) The EMSA of 5'-NTG PAM with N102A Cascade” (Supplementary information, Page 6).

Sup Figure 5: Suggest showing the N and O groups involved in H-bonds.

Response: This figure has been updated to Supplementary Figure 6, and we have included a description explaining that the dashes represent distances, not hydrogen bonds “*Modelling of alternative base pairs at PAM -2 suggests that G_{NT-2} and A_{NT-2} would cause steric clashes with P156 of Cas8b. (A) interaction pattern of Cas8b and PAM-2 in our modelled structure. (B) the initial T_{NT-A_T} pair is substituted as C_{NT-G_T} . The yellow dashes indicate the distances between C_γ of P156 and carbonyl oxygen of the substituted C_{NT-2} . (C) the initial T_{NT-A_T} pair is substituted as A_{NT-T_T} . The*

yellow dash means the distance between C_γ of P156 and N2/N3 of the substituted A_{NT-2} . (D) the initial $T_{NT}-A_T$ pair is substituted as $G_{NT}-C_T$. The yellow dashes indicate the distance between C_β/C_γ of P156 and the amino group of the substituted G_{NT-2} " (Supplementary information, Page 7).

Reviewers' Comments:

Reviewer #1:

Remarks to the Author:

The authors have addressed all my concerns. I have no further comments. I suggest publishing it without delay

Reviewer #2:

Remarks to the Author:

Quantification of DNA binding is an improvement to the manuscript. To evaluate whether one PAM is different to another, each Kd requires the error reported (e.g. standard deviation) and statistical analysis. Values should be adjusted to the appropriate number of significant figures (i.e. likely not to 2 decimal places).

I have concerns about how the quantification was done. Not all of the representative gels match the quantification values in the graphs. For example, ACT has a Kd of 81 nM and ACG has a Kd of 26 nM, yet their gel shifts are remarkably similar (look at the free probe in the higher concentration lanes). Further, in Supp Fig 1C, the EMSA shows binding by GCG is only marginally worse than ACG, yet the %Bound is dramatically worse – GCG has near 100% binding at 200 nM and 100 nM Cascade, yet the graph shows approximately 50% and 30%, respectively. While A-Y-G does appear to have the tightest binding, this does not change the observation that I-B Cascade appears to have a broader PAM requirement. The manuscript should reflect this.

The new details for the affinity assay are a useful addition. The methods section needs to be updated to reflect these. It is an unusual approach to use Sanger sequencing to quantify a heterogeneous population, where typically NGS is used. An issue I see with this approach is in Figure 1, where it appears a **single** blue C peak positioned between the PAM -3 and -2 positions is interpreted as C being the second dominant band for **both** positions. The authors at a minimum need to provide data of their replicates in the supplementary information (indeed also state how many replicates were performed) and do sequencing on the library before Cascade incubation. The starting library is essential as it will demonstrate if there is any initial bias.

Considering the crRNA size being 71 nt, why does Figure 1A say the crRNA is 71-73 nt? Does the type I-B have a variable length spacer? It would be useful to state, or signal on the figure, the length of the type I-B repeat and the spacer.

Point-by-point Response to Referees

Reviewer #2 (Remarks to the Author):

Quantification of DNA binding is an improvement to the manuscript. To evaluate whether one PAM is different to another, each K_d requires the error reported (e.g. standard deviation) and statistical analysis. Values should be adjusted to the appropriate number of significant figures (i.e. likely not to 2 decimal places).

I have concerns about how the quantification was done. Not all of the representative gels match the quantification values in the graphs. For example, ACT has a K_d of 81 nM and ACG has a K_d of 26 nM, yet their gel shifts are remarkably similar (look at the free probe in the higher concentration lanes). Further, in Supp Fig 1C, the EMSA shows binding by GCG is only marginally worse than ACG, yet the %Bound is dramatically worse – GCG has near 100% binding at 200 nM and 100 nM Cascade, yet the graph shows approximately 50% and 30%, respectively.

While A-Y-G does appear to have the tightest binding, this does not change the observation that I-B Cascade appears to have a broader PAM requirement. The manuscript should reflect this.

Response: Thank you for identifying the inaccuracies in our analysis and description of the results. We have re-performed the EMSA experiment for -ANN/NYG- containing target DNA using the Cascade from the same batch. Representative image of the EMSA results for each target DNA and Cascade combination have been displayed in Supplementary Figure 1. Following the methodology described in the Methods section, the binding affinity for each AYN sequence was calculated.

“Electrophoretic mobility shift assay

A final concentration of 10 nM fluorescently labeled target DNA was incubated with titrations of Cascade complex in a 20 μ L total reaction volume containing 20 mM HEPES pH 7.5, 100 mM NaCl. After one hour incubation at 25 $^{\circ}$ C, 10 μ L of each sample was loaded onto 6% acrylamide gel. Electrophoresis was performed in 0.5 \times TBE buffer at 150 V for 35 min in cold room. DNA was visualized by fluorescence imaging in Tanon MINI Space 3000 system and images were quantified using ImageJ software. The fraction of DNA bound (amount of bound DNA divided by the sum of free and bound DNA) was plotted versus the concentration of Cascade and fit to one site-specific binding with Hill slope using GraphPad Prism 8.0.1. Each PAM sequence was tested in at least three independent experiments.” (Line 407-417)

Based on calculated K_d values, we have updated the sentence in the main text to reflect the discovery that the I-B Cascade system is adaptable to a broader spectrum of PAM sequences. “Comparing the binding affinities of DNA sequences containing 5'-A-Y-N-3' with the Cascade complex revealed that AYG has the greatest binding capacity. ATA was a close second, with marginally lower affinity. Both ATY and ACM displayed moderate binding affinities, whereas ACT shows the lowest preference within the AYN group (Supplementary Fig. 1B). The 5'-A-Y-G-3' preference was re-validated by binding assays between Cascade and eight 5'-N-Y-G-3' protospacers (Supplementary Figs. 1C and

1D). The results showed that the nucleotide preference for the -3 position in the PAM sequence was indeed adenine. Taken together, our results validate that the reconstituted Syn type I-B Cascade appears to have a broader PAM requirement with the best preference as 5'-A-Y-G-3'." (Line 125-134)

Supplementary Figure 1. PAM identification by EMSA

The new details for the affinity assay are a useful addition. The methods section needs to be updated to reflect these. It is an unusual approach to use Sanger sequencing to quantify a heterogeneous population, where typically NGS is used. An issue I see with this approach is in Figure 1, where it appears a single blue C peak positioned between the PAM -3 and -2 positions is interpreted as C being the second dominant band for both positions. The authors at a minimum need to provide data of their replicates in the supplementary information (indeed also state how many replicates were performed) and do sequencing on the library before Cascade incubation. The starting library is essential as it will demonstrate if there is any initial bias.

Response: Thank you for pointing out what we overlooked. A detailed methodological description has been incorporated into the main text accordingly. *“The coverage of all 64 PAM sequences was verified by Sanger sequencing to ensure that the DNAs in the library were relatively homogeneous prior to subsequent assays”* (Line 387-389). *“Bands that exhibited specific binding to low concentrations of the Cascade complex were selected, and the extracted DNA from these bands underwent PCR amplification using the primers PF-161 and PR-161, respectively. The PCR products were then analyzed via Sanger sequencing to determine the PAM preference”* (Line 393-397).

We considered that the blue peak indicating C in Figure 1's Sanger sequencing image corresponds to the -2 position, a consistency observed throughout the images in Supplementary Figure 9 (five replicates were performed). Although the Sanger sequencing does not allow for a definitive determination of the preferences at the -2 and -1 positions, the data repeatedly indicate a marked preference for A at the -3 position. Therefore, we began by examining the binding affinity of the Cascade complex with each of the 16 ANNs to determine the nucleotides preferences at positions -2 and -1. After identifying the optimal nucleotides for the -2 and -1 positions, we explored the preference at the -3 position. The results from EMSA experiments strongly supported a significant preference for A at the -3 position, corroborating the insights gained from Sanger sequencing.

Supplementary Figure 9A. Sanger Sequencing images of the constructed PAM library

Supplementary Figure 9B. Sanger sequencing images of PCR-amplified products from Cascade-bound DNA

This method was also applied to validate PAM sequences in the *Tsi* I-A type CRISPR-Cas3 system. The preferences of PAM identified through Sanger sequencing align with those predicted by bioinformatics. (Repurpose Type I-A CRISPR-Cas3 for a Robust HPV Diagnosis. DOI: <https://doi.org/10.21203/rs.3.rs-3354708/v1>)

Considering the crRNA size being 71 nt, why does Figure 1A say the crRNA is 71-73 nt? Does the type I-B have a variable length spacer? It would be useful to state, or signal on the figure, the length of the type I-B repeat and the spacer.

Response: Thank you for the meticulous review. Sequencing data of the *Synechocystis sp.* PCC 6714 I-B CRISPR-Cas system reveal that all repeat sequences are 36 nt in length, whereas spacer sequences vary between 35 and 37 nt, with 35 nt being predominantly observed. Fig. 1A has been updated to show a mature crRNA length of 71 nt, consistent with the 35 nt spacer selected in our experiments.

Figure 1A. Schematic of *Synechocystis sp.* PCC 6714 type I-B operon and mature crRNA

Reviewers' Comments:

Reviewer #2:

Remarks to the Author:

The authors have satisfactorily addressed all my previous concerns. The manuscript is a lot stronger now and is suitable for publication in Nat Comm.